# ADVERSARIAL DRIVING POLICY LEARNING BY MISUNDERSTANDING THE TRAFFIC FLOW

## ABSTRACT

Acquiring driving policies that can transfer to unseen environments is essential for driving in dense traffic flows. Adversarial training is a promising path to improve robustness under disturbances. Most prior works leverage few agents to induce driving policy's failures. However, we argue that directly implementing this training framework into dense traffic flow degrades transferability in unseen environments. In this paper, we propose a novel robust policy training framework that is capable of applying adversarial training based on a coordinated traffic flow. We start by building up a coordinated traffic flow where agents are allowed to communicate Social Value Orientations (SVOs). Adversary emerges when the traffic flow misunderstands the SVO of driving agent. We utilize this property to formulate a minimax optimization problem where the driving policy maximizes its own reward and a spurious adversarial policy minimizes it. Experiments demonstrate that our adversarial training framework significantly improves zero-shot transfer performance of the driving policy in dense traffic flows compared to existing algorithms.

## 1 INTRODUCTION

Policy learning in dense traffic flows is a progressively active area for both academia and industry community in autonomous driving (Dosovitskiy et al., 2017; Suo et al., 2021). Since training driving policy in real world is costly, researchers aim to build dense traffic flows in simulation as an alternative (Cai et al., 2020; Pal et al., 2020; Wu et al., 2021). Peng et al. (2021) develops a traffic flow that exhibits altruistic behaviors and training driving policy in such coordinated flow also performs well. However, the internal dynamics of different traffic flows are varied, making it difficult to train driving policy in one flow and transfer it into unseen traffic patterns. Hence, it is indispensable to develop robust driving policies that can transfer among different traffic flows.

An appealing technical route to improve the robustness of driving policy is adversarial attack (Pinto et al., 2017), which models differences between training and evaluating environments as extra disturbances towards driving policy (Wachi, 2019; Chen et al., 2021; Liu et al., 2021; Huang et al., 2022). To exert disturbances on driving policy, these works leverage few agents to deliberately induce driving policy's failures. Although working well in sparse traffic situations, this pipeline cannot extend to dense traffic flows. On the one hand, increasing the number of attacking agents makes adversarial attacks easier, yet it is harder for the driving policy to resist such strong disturbances, which severely harms policy learning. On the other hand, attacking agents mainly concentrate on producing adversarial behaviors towards driving policy, while overlooking the modeling of altruistic behaviors among them. Therefore, the key is to construct a coordinated traffic flow which still generates adversarial behaviors.

We develop a coordinated traffic flow with communication and propose a misunderstanding-based adversarial training pipeline based on this flow. Specifically, for building a coordinated traffic flow, we introduce the concept of Social Value Orientation (SVO) (Liebrand, 1984) in social psychology which balances egoistic and altruistic behaviors for each agent. SVO can be regarded as the hidden information of one agent, which typically cannot be accessed by other agents. However, in this paper, we allow agents in our traffic flow to communicate genuine SVOs with each other. Since the traffic flow is served as a testbed for training and evaluating driving policies, the coordination mechanism within the traffic flow is invisible to driving policies.

Figure 1: **Overview of our training framework.** *Left*: We build up a coordinated traffic flow in which agents communicate SVOs to coordinate with each other. *Right*: By disturbing the SVO of driving agent, our traffic flow exhibits adversarial behaviors towards the driving policy.

In other words, when placing a driving policy to interact with the traffic flow, the traffic flow requires receiving driving policy's SVO while the driving policy is unaware of traffic flows' SVOs. This property offers a neat approach to induce misunderstandings between driving policy and our traffic flow, making it adversarial towards driving policy. We reserve a spurious adversarial agent to disturb the SVO delivery from the driving agent to other agents and formulate a minimax optimization problem where the driving policy maximizes its own reward while the spurious adversarial policy minimizes it, as shown in Figure 1.

**Contributions.** We propose a novel adversarial training framework based on a coordinated traffic flow to obtain driving policies that can transfer across various traffic flows. We develop a coordinated traffic flow where agents exhibit egoistic, prosocial, and altruistic behaviors based on communicating SVOs with each other. Based on this traffic flow, we apply adversarial driving policy training by adversarially misunderstanding the traffic flow, which is disturbed to produce improper coordinated behaviors towards driving policy. We investigate characteristics of several traffic flows in four challenging scenarios and carry out comprehensive comparative studies to evaluate the robustness of driving policy. Results show that our traffic flow achieves the highest success rate and the proposed adversarial training pipeline significantly improves the transferability of driving policy compared to existing algorithms.

## 2 RELATED WORK

**Dense traffic flows.** Prior works explore different methodologies to simulate dense traffic flows including rule design (Behrisch et al., 2011; Dosovitskiy et al., 2017; Cai et al., 2020; Zhou et al., 2021), Imitation Learning (IL) (Zhao et al., 2021; Gu et al., 2021; Wang et al., 2022), and Multi-Agent Reinforcement Learning (MARL) (Pal et al., 2020; Palanisamy, 2020; Wu et al., 2021). IL naturally leverages numerous human expert data but suffers from severe distribution shift and poor closed-loop performance even in simple scenarios. Most rule- and MARL-based algorithms aim to simulate individual behaviors of distinct agents, which overlooks complex interactions among agents. Similar to our work, Peng et al. (2021) also builds a coordinated traffic flow based on SVO. However, agents in their traffic flow have no access to other agents' SVOs, leading to conservative behaviors.

**Adversarial attack.** A common way to acquire robust policy is applying Robust Adversarial Reinforcement Learning (RARL) (Pinto et al., 2017; Pan et al., 2019; Vinitsky et al., 2020; Oikarinen et al., 2021). Researchers in autonomous driving also follow this pipeline (Wachi, 2019; Ding et al., 2020; Chen et al., 2021; Sharif & Marijan, 2021; Huang et al., 2022). Adversarial policies in Ding et al. (2020); Chen et al. (2021); Sharif & Marijan (2021); Huang et al. (2022) are optimized to collide with driving agent, while Wachi (2019); Huang et al. (2022) attempt to expel ego agent from drivable areas. Using attacking agents to interfere with driving policy deliberately, such pipeline provides large adversarial disturbance for driving policy. However, excessively concerning rarely happened scenarios harms the robustness of driving policy in unseen environments since it fails

to capture simpler yet non-trivial interactive patterns. In this work, we apply adversarial training framework on a coordinated traffic flow with communication to solve this problem.

# 3 TRAFFIC SIMULATION CONSTRUCTION

## 3.1 PROBLEM SETTING

**Partially Observable Stochastic Game (POSG).** Traffic simulation systems are typically formulated as a POSG (Oliehoek & Amato, 2016). Formally, POSG is a tuple $G = \langle \mathcal{I}, \mathcal{S}, \mathcal{A}, P, \mathcal{R}, \rho_0, \mathcal{O}, n, \gamma, T \rangle$. $n$ is the number of agents. $\mathcal{I}$ denotes the set $\{0, 1, \ldots, n-1\}$. $\mathcal{S}$ is the state space. $\mathcal{A}$ is the joint action space of $n$ agents and $\mathcal{A} = \times_{i \in \mathcal{I}} \mathcal{A}_i$. $P : \mathcal{S} \times \mathcal{A} \to \Delta(\mathcal{S})$ [1] is the state transition probability. $\mathcal{R} = \{R_0, R_1, \ldots, R_{n-1}\}$ denotes the set of agent-specific reward functions and $R_i : \mathcal{S} \times \mathcal{A} \to \mathbb{R}$ is bounded for all $i \in \mathcal{I}$. Note that each agent $i$ receives distinct reward from its own reward function $r_i = R_i(s, a)$. $\rho_0 \in \Delta(\mathcal{S})$ is the initial state distribution. $\mathcal{O}$ is the joint observation space and $\mathcal{O} = \times_{i \in \mathcal{I}} \mathcal{O}_i$. $\gamma \in (0, 1]$ is the discount factor, and $T$ is the time horizon. In POSG, each agent $i$ maximizes its own expected cumulative reward via policy $\beta_i : \mathcal{O}_i \to \Delta(\mathcal{A}_i)$. When $n$ becomes large, it is time-and-space consuming to optimize a set of policies $\mathcal{B} = \{\beta_0, \beta_1, \ldots, \beta_{n-1}\}$. To solve this problem, we simply adopt parameter sharing strategy (Terry et al., 2020), i.e., $\beta_i = \beta$, with the help of neural network which has powerful representation ability.

**Incorporating Social Value Orientation (SVO).** From the perspective of social psychology, agents should consider surrounding agents' rewards to achieve coordinated driving. Following Schwarting et al. (2019); Buckman et al. (2019); Peng et al. (2021), we introduce the concept of SVO to model coordinated behaviors among agents and build up coordinated traffic simulation. By incorporating SVO, each agent $i$ maximizes reward with the consideration of other surrounding agents:

$$R_i' = \cos(c_i)R_i + \sin(c_i)R_{S_i} \tag{1}$$

where $R_{S_i} = \sum_{j \in \mathcal{I}_{S_i}} R_j / |\mathcal{I}_{S_i}|$, $\mathcal{I}_{S_i}$ is the set of surrounding agents w.r.t. agent $i$. $c_i \in [0, \frac{\pi}{2}]$ is the SVO of agent $i$ and kept fixed during each episode. Given equation 1, we formulate a SVO-embedded POSG $G' = \langle \mathcal{I}, \mathcal{S}, \mathcal{A}, P, \mathcal{R}', \mathcal{C}, \rho_0, \mathcal{O}, n, \gamma, T \rangle$. $\mathcal{R}' = \{R_0', R_1', \ldots, R_{n-1}'\}$ denotes the set of SVO-embedded reward functions. $\mathcal{C} = \{c_0, c_1, \ldots, c_{n-1}\}$ is set of all SVOs.

**Problem formulation.** As one can see, SVO determines the trade-off between egoistic and altruistic behaviors. For each agent, it is necessary to recognize SVOs of itself and other agents, which provides the ability to infer other agents' reward structures. Therefore, we design policy as $\beta : \mathcal{O}_i \times \mathcal{C}_i \times (\times_{j \in \mathcal{I}_{S_i}} \mathcal{C}_j) \to \Delta(\mathcal{A}_i)$. And we use a single policy $\beta$ to optimize the sum of $n$ optimization objectives in SVO-embedded POSG:

$$\max_{\beta} \mathbb{E}_{s_t \sim P_\beta, a_t \sim \beta} \left[ \sum_{i \in \mathcal{I}} \sum_{t=0}^{T} \gamma^t R_i'(s_t, a_t, c_i) \right], \qquad c_i \in \mathcal{C}_i \tag{2}$$

## 3.2 KEY COMPONENTS

**State space.** Agents driving in dense traffic flow need to continually interact with surrounding agents. Besides, road structures also influence agents' decisions. Therefore, state space $\mathcal{S}$ needs to cover a collection of static and dynamic elements. The set of static elements $E^s$ include lane centerlines, sidelines, agents' global paths, i.e., $E^s = \{centerline, sideline, path\}$. The set of dynamic elements $E^d$ include current and historical poses and velocities (trajectories) of all agents, i.e., $E^d = \{trajectory_0, trajectory_1, \ldots, trajectory_{n-1}\}$. We utilize vectorized representation based on Gao et al. (2020), which is computation- and memory-efficient. In our work, elements in $E^s$ and $E^d$ are sets of points containing corresponding features. Specifically, static element $e_i^s = \{v_0, v_1, \ldots, v_j, \ldots\}, i \in E^s$. $v_j = [p_j, l_i, i, j]$ where $p_j = (x, y, \theta)$ is the pose of point $j$ in element $i$ and $l_i$ is the lane width of element $i$. For points in dynamic elements, $v_j = [p_j, c_i, i, j]$ where $p_j = (x, y, \theta, v)$ and $c_i$ denotes the SVO of agent $i$.

---

[1] $\Delta(\mathcal{X})$ denotes the set of probability distribution over set $\mathcal{X}$.

**Observation space.** In POSG, each agent could only receive perceptual information locally, we use $L_2$ norm to define locality, i.e., agent $i$ could only receive points $(x_e, y_e)$ that $\|(x_i, y_i) - (x_e, y_e)\|_2 \le d$, in which $(x_i, y_i)$ is the current location of agent $i$.

**Design of $\mathcal{R}$.** The goal of each agent in dense traffic flow is homogeneous, for instance, all agents want to successfully finish the task as fast as possible. Besides, since each agent receives $\mathcal{O}_i$, designing $R_i$ upon $\mathcal{O}_i$ rather than $\mathcal{S}$ benefits policy training. In our work, we use self-motivated reward $R_i : \mathcal{O}_i \times \mathcal{A}_i \to \mathbb{R}$ and $R_i = R$ for all $i \in \mathcal{I}$. However, designing self-motivated reward function still remains an open problem. Designing fine-grained dense reward accelerates training procedure but relies too heavily on human knowledge, while training with coarse sparse reward requires much more data. To combine both benefits, we design a near-sparse reward function containing a dense reward for incentive driving fast and a sparse reward for penalizing catastrophic failures. Catastrophic failures include collision with other agents, deviation from drivable area, driving too far from global path, and crashing into wrong lane. Coordinated behaviors could be produced by incorporating SVO.

**Policy training.** We apply Independent Policy Learning (IPL) (Tan, 1993) to solve Equation 2. Although IPL is prone to generate egoistic suboptimal behaviors (de Witt et al., 2020; Yang et al., 2020), we could alleviate this problem by incorporating SVO, which forces the algorithm to consider other agents' goals.

### 3.3 Policy Architecture

To better extract static and dynamic features and capture relations among them, we utilize a hierarchical feature extraction framework. We use DeepSet (Zaheer et al., 2017) to aggregate homogeneous information within dynamic and static elements, followed by Multi-Head Attention (MHA) (Vaswani et al., 2017) to further extract heterogeneous information among different elements.

**Homogeneous feature aggregation.** Consider the elements set $e \subset E, E = \{E^s, E^d\}$, and the function processing on the set needs to retain the adjacency between elements and permutation-invariant to the order of objects in the element. Based on theorem 2 in (Zaheer et al., 2017), the propagation function $f$ is defined as:

$$f(e) = \rho \left( \sum_{v \in e} \phi(v) \right) \tag{3}$$

And we obtain the element level features $l_e = f(e)$, where $e$ is the input elements set, the nodes $v \in e$ transformed into a representation $\phi(v)$. The sum of representations is processed using the $\rho$ network defined by Multi-Layer Perception (MLP) network. In our implementation, DeepSet can extract polyline-level features while not introducing too many parameters.

**Heterogeneous feature aggregation.** The static element level features $l_e^s = [l_{e_0}^s, l_{e_1}^s, \ldots, l_{e_j}^s, \ldots]$ and the dynamic element features $l_e^d = [l_{e_0}^d, l_{e_1}^d, \ldots, l_{e_j}^d, \ldots]$ go through a MHA layer which takes into account their inter-relations to output the final action for the agents. Given arbitrary feature matrices $w, z$ and their linear projections $w_Q, w_K, w_V$ and $z_Q, z_k, z_v$, the $SelfAttn(w)$ and $CrossAttn(w, z)$ are defined as:

$$SelfAttn(w) = \frac{Softmax\left(w_Q w_K^T\right)}{\sqrt{d_k}} w_V$$

$$CrossAttn(w, z) = \frac{Softmax\left(w_Q z_K^T\right)}{\sqrt{d_k}} z_V \tag{4}$$

where $\sqrt{d_k}$ is the dimension of the key vectors. We leverage the one-layer cross-attention network to model the interaction between dynamic and static segments. The dynamic elements features $l_e^d$ and static elements features $l_e^s$ are fused by the $SelfAttn$ and $CrossAttn$ operation:

$$l_o^d = SelfAttn(l_e^d) + CrossAttn(l_e^d, l_e^s)$$

$$l_o^s = SelfAttn(l_e^s) + CrossAttn(l_e^s, l_e^d) \tag{5}$$

$l_o = \{l_o^d, l_o^s\}$ is the final output of MHA. We then decode the agents' action from $l_o$:

$$a = \varphi(l_o) \tag{6}$$

where $\varphi(\cdot)$ is the action decoder, and $a \in \mathcal{A}$. For simplicity, we use an MLP as the decoder function.

---

**Algorithm 1:** Misunderstanding-based Adversarial Reinforcement Learning

---

**Input:** SVO-embedded POMDP $M$ containing traffic flow policy $\beta$
**Output:** Driving policy $\pi$, adversarial policy $\pi_c$
**Initialize:** Learnable parameters $\theta_\pi, \theta_{\pi_c}$
**for** $n = 1, 2, \ldots, N$ **do**
    /* Stage 1: Given $\pi_c$ optimize $\pi$                                 */
    **for** $n_1 = 1, 2, \ldots, N_1$ **do**
        Collect a set of transition tuples $\{(o, a, o', r)\}$ trajectories by rolling out $\pi$ and $\pi_c$ on $M$;
        Optimize parameters $\theta_\pi$ of $\pi$ using any RL algorithms;
    /* Stage 2: Given $\pi$ optimize $\pi_c$                                 */
    **for** $n_2 = 1, 2, \ldots, N_2$ **do**
        Collect a set of transition tuples $\{(o, a, c_\beta, o', -r)\}$ trajectories by rolling out $\pi$ and $\pi_c$ on $M$;
        Optimize parameters $\theta_{\pi_c}$ of $\pi_c$ using any RL algorithms;

---

# 4 MISUNDERSTANDING-BASED ADVERSARIAL LEARNING

## 4.1 PROBLEM SETTING

For single-agent driving task, we formulate SVO-embedded Partially Observable Markov Decision Process (POMDP) as $M = \langle \mathcal{S}, \mathcal{A}, P, \mathcal{R}, \mathcal{C}, \rho_0, \mathcal{O}, \gamma, T, \beta \rangle$. Note that $\beta$ is the policy that controls the traffic simulation and affects state transition probability $P$. Following the tradition, we define policy $\pi : \mathcal{O} \to \Delta(\mathcal{A})$ to solve $M$. $c_\beta \in \mathcal{C} = [0, \frac{\pi}{2}]$ is the SVO of driving policy which is taken by $\beta$. The genuine SVO of driving agent $c_\pi$ is always 0 since existing single-agent algorithms are fully self-interested. Adversary emerges when $c_\beta$ and $c_\pi$ differ:

$$\max_\pi \min_{c_\beta} \mathbb{E}_{s_t \sim P_{\beta, c_\beta}, a_t \sim \pi}[\sum_{t=0}^{T} \gamma^t R(s_t, a_t)] \tag{7}$$

In section 3.1 and equation 7, the SVOs are invariant during one episode for the reason of stabilizing training. However, in adversarial training, we aim to destabilize policy training. Therefore, we introduce a spurious policy $\pi_c : \mathcal{O} \to \Delta(\mathcal{C})$ to produce $c_\beta$ which is allowed to change across time steps, changing equation 7 into:

$$\max_\pi \min_{\pi_c} \mathbb{E}_{c_{\beta,t} \sim \pi_c, s_t \sim P_{\beta, c_{\beta,t}}, a_t \sim \pi}[\sum_{t=0}^{T} \gamma^t R(s_t, a_t)] \tag{8}$$

Note that equation 8 relates to three policies including driving policy $\pi$, background policy $\beta$, and spurious adversarial policy $\pi_c$. Since driving policy maximizes its own reward, it is egoistic from the perspective of social psychology. Background policy controls the whole traffic flow to exhibit egoistic and altruistic behaviors. The spurious policy is the only one that aims to generate adversarial behaviors by minimizing driving policy's reward.

We highlight that agents in our traffic flow try to coordinate with each other, which is a fundamental difference compared to previous attacking agents. Instead of deliberately inducing failures of $\pi$, we keep $\beta$ non-adversarial and leverage an extra $\pi_c$ to disturb the SVO of $\pi$ taken by $\beta$.

## 4.2 ADVERSARIAL POLICY TRAINING

Algorithm 1 outlines our training framework. Given background policy $\beta$, we alternatively optimize both driving policy $\pi$ and adversarial policy $\pi_c$. The parameters $\theta_\pi$ of $\pi$ and $\theta_{\pi_c}$ of $\pi_c$ are randomly initialized before training. In each of $N$ iterations, we first optimize $\theta_\pi$ and keep $\theta_{\pi_c}$ fixed, followed by optimizing $\theta_{\pi_c}$ and keeping $\theta_\pi$ fixed.

In Stage 1, we iterate $N_1$ times to optimize driving policy $\pi$. By sampling POMDP $M$, we collect a set transition tuples $\{(o, a, o', r)\}$, where $o, o' \in \mathcal{O}$, $a \in \mathcal{A}$, and $r = R(o, a)$. We then apply standard RL algorithms to optimize $\theta_\pi$. In Stage 2, we iterate $N_2$ times to optimize adversary policy $\pi_c$. Similar to Stage 1, we sample $M$ and get another set of transition tuples and apply RL algorithms. Differently, we reverse the sign of $r$ since adversary policy aims to decrease the

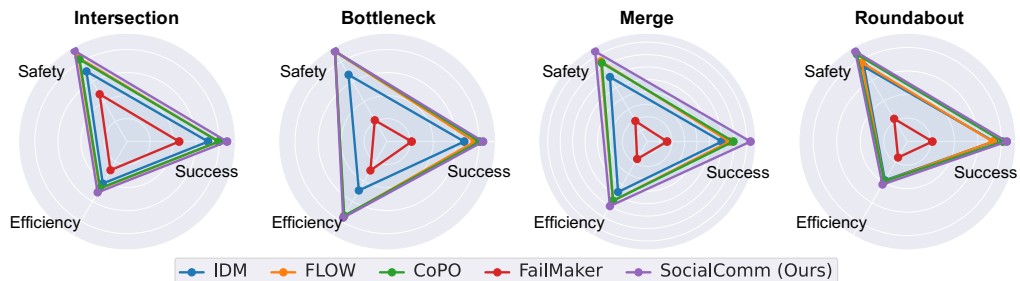

Figure 2: **Performance of different traffic flows.** The radar graphs demonstrate three essential features of different traffic flows. Safety is calculated by taking the complement of catastrophic failures.

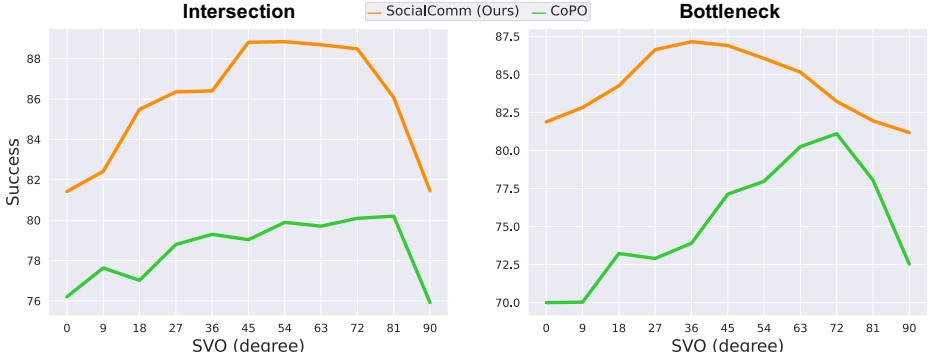

Figure 3: **Success rates of CoPO and SocialComm (Ours).** The figure reports the percentage of success rates in `intersection` and `bottleneck`. We assign a fixed SVO from $0°$ to $90°$ at regular intervals. All agents in traffic flows are given the same SVO.

performance of driving policy. Note that the action of adversarial policy is $c_\beta$. $a$ is the action of driving policy which is used to compute $r$. This alternating procedure is repeated for $N$ iterations.

The main difference between standard and misunderstanding-based adversarial learning is the objective of $\beta$. In standard adversarial learning, $\beta$ controls background agents to attack the driving agent. Background agents know exactly which one is the driving agent. While in misunderstanding-based adversarial learning, background agents aim to coordinate with each other, including the driving agent. A background agent cannot distinguish which surrounding agent is the driving agent. Therefore, the spurious agent which produces $c_\beta$ applies adversary from the perspective of driving agent. In Algorithm 1 the spurious agent and driving policy take the same observation.

## 5 RESULTS

In this section, we pursue to answer three seminal questions. (1) Can our proposed traffic simulation produce more coordinated behaviors? (2) Does the spurious adversarial policy degrade driving policy's performance? (3) Does our adversarial training framework improve driving policy's zero-shot transfer ability? Before discussing these questions, we first explain some preliminary details.

**Settings.** We evaluate our proposed method using our internal driving simulator which supports various maps and scenarios. Similar to Peng et al. (2021), we select several highly interactive scenarios including `intersection`, `bottleneck`, `merge`, and `roundabout`. During training, we randomly place 8 to 20 vehicles in each scenario at the beginning of each episode. After training, we randomly place 20 vehicles and evaluate all relevant methods. See more details of our simulator in Appendix A.4.

**Metrics.** We consider three kinds of widely-accepted and general metrics. Firstly, success rate of the whole traffic simulation (Success). Secondly, catastrophic failure rates of the whole traffic simulation. Catastrophic failures include collision between agents (Collision), deviation from drivable area (Off Road), driving too far from global path (Off Route), and crashing into wrong lane (Wrong

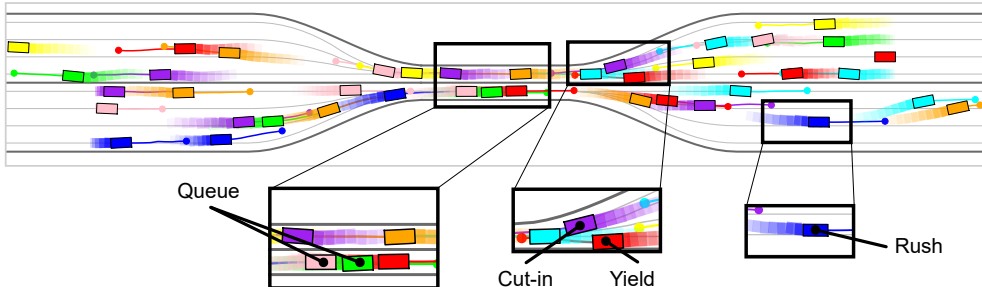

Figure 4: **Coordinated behaviors in `bottleneck`.** The figure highlights that our traffic flow with communication produces diverse coordinated behaviors such as queueing at the narrow crossing, rushing at open areas and yielding to avoid crashes.

Lane). Third, driving efficiency is represented by average speed of the whole traffic simulation (Efficiency). As for single-agent training, these metrics are calculated from the perspective of ego agent. More discussions can be found in Appendix A.5.

**Traffic flows.** We denote our traffic flow as SocialComm and implement four representative traffic flows to carry out comparative studies on different traffic flows and training pipelines. (1) Intelligent Driver Model (IDM) (Treiber & Kesting, 2013) is a rule-based controller which uses one single differential equation to model longitudinal movements for all agents. Each agent strictly follows its global path. (2) FLOW (Wu et al., 2021) is a MARL-based method where each agent aims to maximize its own reward. (3) CoPO (Peng et al., 2021) is a MARL-based method that also incorporates SVO. CoPO has two stages. First, similar to our traffic flow, CoPO applies IPL and SVO to train a background policy. Then, CoPO additionally trains a meta-controller to select all agents' SVOs so that the success rate of the whole population is maximized. Note that agents in CoPO have no access to other agents' SVOs. (4) FailMaker (Wachi, 2019) is a method to generate attacking behaviors. Attacking agents are rewarded if they successfully induce driving agent' catastrophic failures while avoiding personal failures except for collision.

**Training pipelines.** We use vanilla RL (VRL), existing robust adversarial RL (RARL), and our proposed misunderstanding-based adversarial learning (M-RARL) to train driving policies. VRL can be applied in IDM, FLOW, CoPO, and SocialComm. RARL and M-RARL can only be applied in FailMaker and SocialComm (with the spurious adversarial agent) respectively.

## 5.1 PERFORMANCE OF TRAFFIC FLOWS

We demonstrate the performance of different traffic flows. Figure 2 and Figure 3 show quantitative results of different traffic flows. As one can see, our proposed SocialComm achieves the highest success rates and average speeds across all scenarios. Compared with CoPO, agents in SocialComm can recognize other agents' SVOs and produce coordinated behaviors, therefore achieving collaboration and high efficiency of the whole system. FailMaker achieves the lowest success rate and highest collision rate (lowest safety) due to its adversarial nature. Note that in `merge`, our traffic flow outperforms other methods by a large margin. The reason is that the initial poses of all agents in `merge` are much closer than these in other scenarios, which makes it harder for the agents to coordinate. Qualitative results are shown in Figure 4. See Appendix A.2 for more results.

## 5.2 MISUNDERSTANDING-BASED ADVERSARY WITH SOCIALCOMM

Our robust policy learning framework applies adversarial training on a coordinated traffic flow. In this part, we demonstrate that misunderstanding-based adversary successfully degrades driving policy's performance and has the ability to impede the driving agent.

**Performance.** We first train driving policy using vanilla RL in four non-adversarial traffic flows (including IDM, FLOW, CoPO, and SocialComm) and deploy these well-trained driving policies into SocialComm with the spurious adversarial agent. Results are shown in Table 1. Data in parentheses is the change of performances under the spurious adversarial policy. Success rates and efficiencies of all driving policies decrease. This reveals that our spurious adversarial policy could impede the

Table 1: **Effect of misunderstanding-based adversary with our coordinated traffic flow SocialComm.** The table reports the percentage of different metrics in `intersection` and `bottleneck`. Results in parentheses indicate the performance change under adversary. Results marked in red indicate the performance degradation under adversary while results in blue indicate the performance increase. A "†" indicates our proposed traffic flow.

| Methods | Intersection | | | | | |
| --- | --- | --- | --- | --- | --- | --- |
| | Success ($\uparrow$) | Collision ($\downarrow$) | Off Road ($\downarrow$) | Off Route ($\downarrow$) | Wrong Lane ($\downarrow$) | Efficiency ($\uparrow$) |
| VRL/IDM | 77.0 (-2.5) | 10.0 (+4.5) | 11.0 (-3.5) | 1.0 (+1.5) | 0.0 (+0.0) | 47.1 (-1.2) |
| VRL/FLOW | 84.0 (-1.5) | 13.5 (+1.5) | 1.0 (+1.0) | 0.5 (-0.5) | 0.5 (-0.5) | 50.1 (-0.5) |
| VRL/CoPO | 81.5 (-3.0) | 14.5 (+4.0) | 1.0 (+0.5) | 2.0 (-1.0) | 0.0 (+0.0) | 48.8 (-1.3) |
| VRL/SocialComm† | 87.0 (-2.0) | 7.0 (+2.0) | 1.5 (+1.5) | 1.5 (-1.0) | 0.5 (+0.0) | 51.9 (-1.1) |

| Methods | Bottleneck | | | | | |
| --- | --- | --- | --- | --- | --- | --- |
| | Success ($\uparrow$) | Collision ($\downarrow$) | Off Road ($\downarrow$) | Off Route ($\downarrow$) | Wrong Lane ($\downarrow$) | Efficiency ($\uparrow$) |
| VRL/IDM | 52.5 (-2.0) | 26.0 (+1.0) | 21.0 (+1.0) | 0.5 (+0.0) | 0.0 (+0.0) | 58.3 (-1.3) |
| VRL/FLOW | 75.5 (-2.0) | 19.5 (+4.5) | 4.5 (-2.5) | 0.5 (+0.0) | 0.0 (+0.0) | 74.6 (-0.8) |
| VRL/CoPO | 71.5 (-2.0) | 13.5 (-2.5) | 15.5 (+3.5) | 1.0 (+0.0) | 0.0 (+0.0) | 70.9 (-1.0) |
| VRL/SocialComm† | 91.0 (-6.0) | 4.0 (+3.0) | 5.0 (+3.0) | 0.0 (+0.0) | 0.0 (+0.0) | 81.3 (-3.0) |

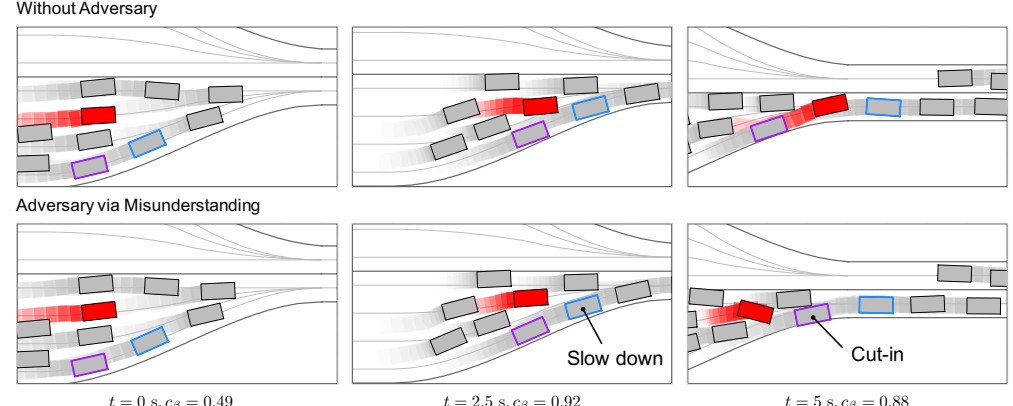

Figure 5: **Adversarial behaviors towards driving agent generated by coordinated traffic flow.** Driving policy controls the red vehicle. Vehicles with blue and purple boxes are two background agents that exhibit adversarial behaviors towards driving policy while maintaining coordination.

driving policy. Note that catastrophic failures of driving policy still increase under adversary in our traffic flow. The reason is that although highly coordinated, our traffic flow cannot eliminate catastrophic failures of the whole traffic system. And it is not unallowable for the traffic flow to incur driving policy's catastrophic failures due to the optimization objective in Equation 8.

**Adversarial behaviors on coordinated traffic flow.** Figure 5 demonstrates some qualitative results on how our traffic flow impedes driving policy to finish its own task. Driving policy controls the red car. At $t = 0$s, the driving policy aims to pass through the bottleneck efficiently and keeps high speed. When driving policy approaches the bottleneck, where interaction frequently happens, the agent with blue box slows down ($t = 2.5$s) to regulate the speed of driving policy and pass through the bottleneck. After that, the agent with purple box cut in the agent ahead of driving agent ($t = 5.0$s). These agents impede the driving policy and degrade its efficiency and explicit coordinated behaviors among each other.

## 5.3 ZERO-SHOT TRANSFER PERFORMANCE OF DRIVING POLICIES

To evaluate the robustness in unseen environments, we deploy all driving policies in all traffic flows. In Figure 6, elements in primary diagonals are obtained by evaluating in their training environments (the traffic flow used in training and evaluating is the same) and typically achieve highest success rates. Therefore, off-diagonal elements reveal zero-shot transfer performances and colors in the

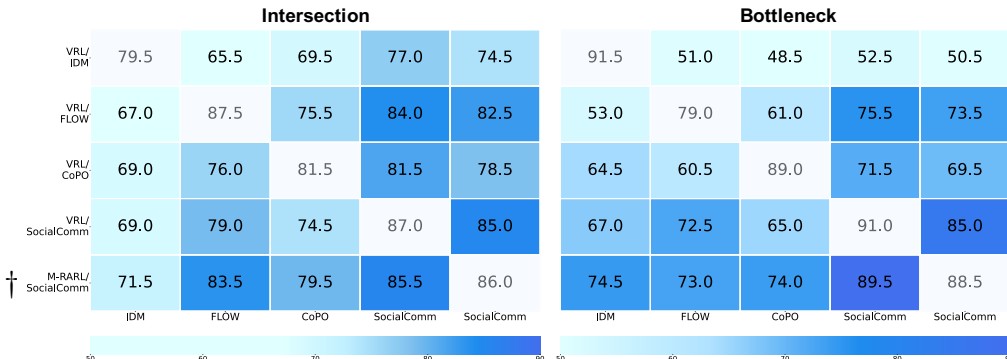

Figure 6: **Zero-shot transfer performance in `intersection` and `bottleneck`.** The heatmap reports the percentage of success rate for different methods in different traffic flows. Deeper color represents higher success rate. Primary diagonals indicates that training and evaluating environments are the same. A "†" indicates our proposed misunderstanding-based adversarial training.

lowest row are column-wise deepest, indicating that driving policy trained with our proposed M-RARL acquires highest transferability.

RARL/FailMaker shows the worst zero-shot transfer performance (which is removed from Figure 6 for clarity and can be found in Appendix A.3) since background agents in FailMaker deliberately induce catastrophic failures of driving policy. This means that RARL/FailMaker has no way to see non-adversarial traffic behaviors. Therefore, although robustness under adversarial attack is improved, RARL/FailMaker is fragile to unseen traffic patterns. Based on this observation, VRL/IDM, VRL/FLOW, VRL/CoPO, and VRL/SocialComm demonstrate superior zero-shot transfer performances in non-adversarial traffic flows compared to RARL/FailMaker.

Comparing VRL/SocialComm and M-RARL/SocialComm, one can see that injecting adversaries properly in dense traffic flow significantly improves robustness in unseen non-adversarial environments. Note that all methods except RARL/FailMaker act poorly in FailMaker since it is extremely easy for background agents in FailMaker to attack driving policies, no matter how driving agents act shrewdly. See Appendix A.3 for more results.

## 6    CONCLUSION

In this paper, we propose a novel adversarial training framework based on a coordinated traffic flow with communication. Driving policies trained with this framework exhibit robust behaviors across various traffic flows. We report characteristics of several traffic flows in scenarios including `intersection`, `bottleneck`, `merge`, and `roundabout`. We carry out numerous comparative studies to evaluate the transferability of driving policy. Results show that our traffic flow achieves the highest success rate and adversarial learning on our traffic flow significantly improves driving policy's zero-shot transfer performance compared to existing algorithms.

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

# A Appendix

## A.1 Implementation Details

**Intelligent Driver Model (IDM).** IDM is given by Equation 9 and Equation 10. The model describes the acceleration $\dot{v}_{back}$ of the back agent, as a function of the agent's velocity $v_{back}$, the reference velocity $v_0$, the difference between the agent velocity and the velocity of the agent in front $\Delta v = v_{back} - v_{front}$, and the following distance $\varphi = s_{front} + L_{length,front} - s_{back}$. Here, $s_{front}$ is the position of the front agent, $s_{back}$ denotes the position of the back agent, and $L_{length,front}$ denotes the length of the front agent. The physical interpretation of the parameters are the minimum following time, $T$, the minimum following gap, $s_0$, the maximum acceleration, $a$, the minimum following gap, $s_0$, the maximum acceleration, $a$, and the comfortable braking deceleration, b.

$$\dot{v}_{back} = a\left[1 - \left(\frac{v_{back}}{v_0}\right)^{\delta} - \left(\frac{\phi\left(v_{back}, \Delta v\right)}{\varphi}\right)^2\right] \tag{9}$$

$$\phi\left(v_{back}, \Delta v\right) = s_0 + v_{back}T + \frac{v_{back}\Delta v}{2\sqrt{ab}} \tag{10}$$

**Policy learning parameters.** For Independent Policy Learning (IPL) and single-agent reinforcement learning algorithms, we utilize Soft-Actor-Critic (SAC) (Haarnoja et al., 2018) and Adam optimizer (Kingma & Ba, 2015). Detailed parameters are shown in Table 3.

## A.2 Results on Our Coordinated Traffic Flow

More results are shown in Table 4, Figure 7, Figure 8, Figure 9, Figure 10, Figure 11, Figure 12, Figure 13, and Figure 14.

## A.3 Results on Zero-shot Transfer

More results are shown in Table 5, Table 6 Table 7, Table 8.

## A.4 Details of our Simulator

Our internal driving simulator is 2D and aims to investigate single- and multi-agent driving behaviors, especially in dense traffic flows. Inspired by the trajectory prediction community (Liang et al., 2020; Zhao et al., 2021; Gu et al., 2021), our simulator utilizes sparse (vectorized) representation to capture the structural information of high-definition maps and agents. Compared to rasterized encoding which rasterizes the HD map elements together with agents into an image, vectorized representation is computation- and memory-efficient (Gao et al., 2020). The critical components of our simulator are built on top of this vectorized representation. For designing an RL-oriented simulator, there are three critical components including scenario initialization, step forward, and done condition.

**scenario initialization.** We choose one scenario from `intersection`, `bottleneck`, `merge`, and `roundabout` and load the pre-built vectorized map for this scenario. After that, we assign global path, initial pose, and SVO for each agent in the scenario. A vectorized map contains two parts including the centerline and sideline. Each part is a 3D tensor which contains different elements. Each element contains a sequence of points. Each point $v$ is a vector $[p, l]$ where $p = (x, y, \theta)$ is the pose and $l$ represents the lane width (for sideline, $l$ is always 0). The average distance of adjacent points is $2.0m$. Given this map, we build a graph $\mathcal{G}$ on top of centerline where each element is a node of $\mathcal{G}$ and an edge exists only when two elements are connected end to end. For each scenario, we manually pick up two bunch of points as initial and terminational poses respectively. For each agent, we randomly select an initial and terminal pose ($p_{initial}$ and $p_{terminal}$) and use A* algorithm to search a list of points from $p_{initial}$ to $p_{terminal}$ on $\mathcal{G}$. This list of points is the global path of the agent, in which the first point is the initial pose. Finally, we assign a SVO $c \in [0, \frac{\pi}{2}]$ to this agent. For all agents in the scenario, the above procedure is repeated. We further clarify what the term "agent" means in our paper. Agents can be divided into two categories: foreground agents and background agents. Background agents are NPCs that have unchanged policies like IDM and

learned NNs. Background agents are part of the environment. For a single-agent environment, there is only one single foreground agent (driving agent). For a multi-agent environment, there exist multiple foreground agents. "Foreground agent" is exactly the meaning of "agent" in RL community. Currently, when there are $n$ vehicles in our simulator, the number of foreground and background agents is $(1, n-1)$ for single-agent settings and $(n, 0)$ for multi-agent settings.

**step forward.** We use bicycle model as the vehicle dynamic model, where the inputs of the model are acceleration $a$ and steer $\delta$ and the state is $(x, y, \theta, v)$. To guarantee that the agent will not exceed its maximum speed $v_{max} = 6m/s$ substantially, we introduce a PID controller ($K_p = 1.0$, $K_i = 0.01$, $K_d = 0.05$) to regulate $a$ given the reference speed $v_r$ and current speed $v$. Therefore, the action is $(v_r, \delta)$ where $v_r \in [0, v_{max}]$ and $\delta \in [-45°, 45°]$. As explained in Section 3.2, the state and observation space contains a collection of static and dynamic elements and is vectorized, the dimension of state and observation space is inherently not fixed. The length of static elements is not fixed and has no upper bound. The upper bound of dynamic elements' length is 10.

**done condition.** In our simulator, one agent is done if it reaches its destination, encounters catastrophic failures, or survives until timeout. Once an agent is done, it will be removed from the scenario. When all foreground agents are done, this episode ends. Catastrophic failures include collision with other agents, deviation from drivable area, driving too far from global path, and crashing into wrong lane. The maximum steps for one episode are $t_{max}$ and when an agent survives $t_{max}$ steps in the environment, we call it "timeout". In this paper, $t_{max} = 100$. An agent is marked as success only when it passes the interaction zone (as shown in Figure 15). Note that we name each scenario with its interaction zone. For instance, the interaction zone in `bottleneck` is the bottleneck.

## A.5 METRICS

Note that most metrics we use are widely used in prior works. Success is used in Dosovitskiy et al. (2017); Chen et al. (2020); Wu et al. (2021); Peng et al. (2021); Rhinehart et al. (2019); Chen et al. (2019b); Cai et al. (2019). Collision is used in Chen et al. (2020); Suo et al. (2021); Chen et al. (2019b); Cai et al. (2019). Off Road is used in Rhinehart et al. (2019); Chen et al. (2019b). Off Route is our design, but Chen et al. (2019a); Toromanoff et al. (2020) uses it as a reward term. Wrong Lane is used in Rhinehart et al. (2019). Efficiency is used in Wu et al. (2021); Chen et al. (2019b); Moghadam et al. (2020); Cai et al. (2019).

## A.6 ADDITIONAL RESULTS

The comparison of training efficiency between CoPO and SocialComm (Ours) is shown in Figure 16.

The impact of varying the number of agents in each scenario is shown in Figure 17.

Table 2: **Parameters of IDM.**

| Parameter | Value |
|---|---|
| Desired speed $v_0$ | $6\ m/s$ |
| Time gap $T$ | $1.0\ s$ |
| Minimum gap $s_0$ | $2\ m$ |
| Acceleration exponent $\delta$ | 4 |
| Acceleration $a$ | $5.0\ m/s^2$ |
| Comfortable deceleration $b$ | $5.0\ m/s^2$ |

Table 3: **Hyperparameters of SAC.**

| Parameter | Value |
|---|---|
| optimizer | Adam |
| actor learning rate | $1 \cdot 10^{-4}$ |
| critic learning rate | $5 \cdot 10^{-4}$ |
| tune learning rate | $1 \cdot 10^{-4}$ |
| discount ($\gamma$) | 0.9 |
| batch size | 128 |
| replay buffer size | $10^6$ |
| nonlinearity | ReLU |
| target smoothing coefficient ($\tau$) | 0.005 |
| target update interval | 200 |

Table 4: **Quantitative performance of traffic flows.** The table reports the percentage of different metrics in `intersection`, `bottleneck`, `merge`, and `roundabout`. A "†" indicates our proposed traffic flow.

| Traffic Flows | Intersection | | | | | |
|---|---|---|---|---|---|---|
| | Success (↑) | Collision (↓) | Off Road (↓) | Off Route (↓) | Wrong Lane (↓) | Efficiency (↑) |
| IDM | $70.4 \pm 0.0$ | $29.6 \pm 0.0$ | $\mathbf{0.0 \pm 0.0}$ | $\mathbf{0.0 \pm 0.0}$ | $\mathbf{0.0 \pm 0.0}$ | $42.1 \pm 0.0$ |
| FLOW | $79.6 \pm 0.5$ | $15.9 \pm 0.6$ | $2.0 \pm 0.2$ | $1.2 \pm 0.1$ | $\mathbf{0.0 \pm 0.0}$ | $47.4 \pm 0.2$ |
| CoPO | $79.6 \pm 0.3$ | $17.6 \pm 0.4$ | $1.3 \pm 0.1$ | $1.4 \pm 0.1$ | $0.1 \pm 0.1$ | $46.8 \pm 0.1$ |
| FailMaker | $45.3 \pm 0.3$ | $52.5 \pm 0.3$ | $\mathbf{0.0 \pm 0.0}$ | $\mathbf{0.0 \pm 0.0}$ | $\mathbf{0.0 \pm 0.0}$ | $28.7 \pm 0.1$ |
| SocialComm† | $\mathbf{86.9 \pm 0.5}$ | $\mathbf{9.0 \pm 0.4}$ | $2.8 \pm 0.1$ | $1.2 \pm 0.1$ | $0.2 \pm 0.1$ | $\mathbf{51.0 \pm 0.2}$ |

| Traffic Flows | Bottleneck | | | | | |
|---|---|---|---|---|---|---|
| | Success (↑) | Collision (↓) | Off Road (↓) | Off Route (↓) | Wrong Lane (↓) | Efficiency (↑) |
| IDM | $67.0 \pm 0.0$ | $33.0 \pm 0.0$ | $\mathbf{0.0 \pm 0.0}$ | $\mathbf{0.0 \pm 0.0}$ | $\mathbf{0.0 \pm 0.0}$ | $49.1 \pm 0.0$ |
| FLOW | $76.2 \pm 0.5$ | $9.8 \pm 0.4$ | $14.3 \pm 0.6$ | $\mathbf{0.0 \pm 0.0}$ | $\mathbf{0.0 \pm 0.0}$ | $75.1 \pm 0.3$ |
| CoPO | $80.3 \pm 0.6$ | $\mathbf{9.3 \pm 0.7}$ | $11.4 \pm 0.4$ | $\mathbf{0.0 \pm 0.0}$ | $\mathbf{0.0 \pm 0.0}$ | $74.5 \pm 0.3$ |
| FailMaker | $21.3 \pm 0.2$ | $78.6 \pm 0.1$ | $0.1 \pm 0.0$ | $\mathbf{0.0 \pm 0.0}$ | $\mathbf{0.0 \pm 0.0}$ | $29.0 \pm 0.2$ |
| SocialComm† | $\mathbf{83.4 \pm 0.4}$ | $9.4 \pm 0.3$ | $7.3 \pm 0.3$ | $\mathbf{0.0 \pm 0.0}$ | $\mathbf{0.0 \pm 0.0}$ | $\mathbf{76.4 \pm 0.1}$ |

| Traffic Flows | Merge | | | | | |
|---|---|---|---|---|---|---|
| | Success (↑) | Collision (↓) | Off Road (↓) | Off Route (↓) | Wrong Lane (↓) | Efficiency (↑) |
| IDM | $60.0 \pm 0.0$ | $40.0 \pm 0.0$ | $\mathbf{0.0 \pm 0.0}$ | $\mathbf{0.0 \pm 0.0}$ | $\mathbf{0.0 \pm 0.0}$ | $46.9 \pm 0.0$ |
| FLOW | $66.2 \pm 0.4$ | $25.4 \pm 0.5$ | $8.5 \pm 0.2$ | $\mathbf{0.0 \pm 0.0}$ | $\mathbf{0.0 \pm 0.0}$ | $55.1 \pm 0.2$ |
| CoPO | $69.3 \pm 0.5$ | $26.9 \pm 0.6$ | $3.8 \pm 0.3$ | $\mathbf{0.0 \pm 0.0}$ | $\mathbf{0.0 \pm 0.0}$ | $54.9 \pm 0.2$ |
| FailMaker | $16.0 \pm 0.2$ | $80.9 \pm 0.2$ | $3.4 \pm 0.2$ | $\mathbf{0.0 \pm 0.0}$ | $\mathbf{0.0 \pm 0.0}$ | $16.2 \pm 0.1$ |
| SocialComm† | $\mathbf{83.1 \pm 0.5}$ | $\mathbf{16.2 \pm 0.5}$ | $0.6 \pm 0.1$ | $0.2 \pm 0.1$ | $\mathbf{0.0 \pm 0.0}$ | $\mathbf{60.0 \pm 0.2}$ |

| Traffic Flows | Roundabout | | | | | |
|---|---|---|---|---|---|---|
| | Success (↑) | Collision (↓) | Off Road (↓) | Off Route (↓) | Wrong Lane (↓) | Efficiency (↑) |
| IDM | $73.6 \pm 0.0$ | $26.4 \pm 0.0$ | $\mathbf{0.0 \pm 0.0}$ | $\mathbf{0.0 \pm 0.0}$ | $\mathbf{0.0 \pm 0.0}$ | $38.1 \pm 0.0$ |
| FLOW | $72.7 \pm 0.6$ | $22.4 \pm 0.4$ | $4.8 \pm 0.2$ | $\mathbf{0.0 \pm 0.0}$ | $0.1 \pm 0.1$ | $39.2 \pm 0.1$ |
| CoPO | $81.2 \pm 0.6$ | $14.3 \pm 0.5$ | $4.0 \pm 0.2$ | $\mathbf{0.0 \pm 0.0}$ | $0.5 \pm 0.1$ | $39.0 \pm 0.1$ |
| FailMaker | $21.3 \pm 0.3$ | $77.6 \pm 0.3$ | $0.5 \pm 0.0$ | $\mathbf{0.0 \pm 0.0}$ | $0.6 \pm 0.1$ | $15.7 \pm 0.1$ |
| SocialComm† | $\mathbf{84.6 \pm 0.5}$ | $\mathbf{11.5 \pm 0.3}$ | $3.6 \pm 0.3$ | $0.1 \pm 0.0$ | $0.5 \pm 0.1$ | $\mathbf{42.2 \pm 0.1}$ |

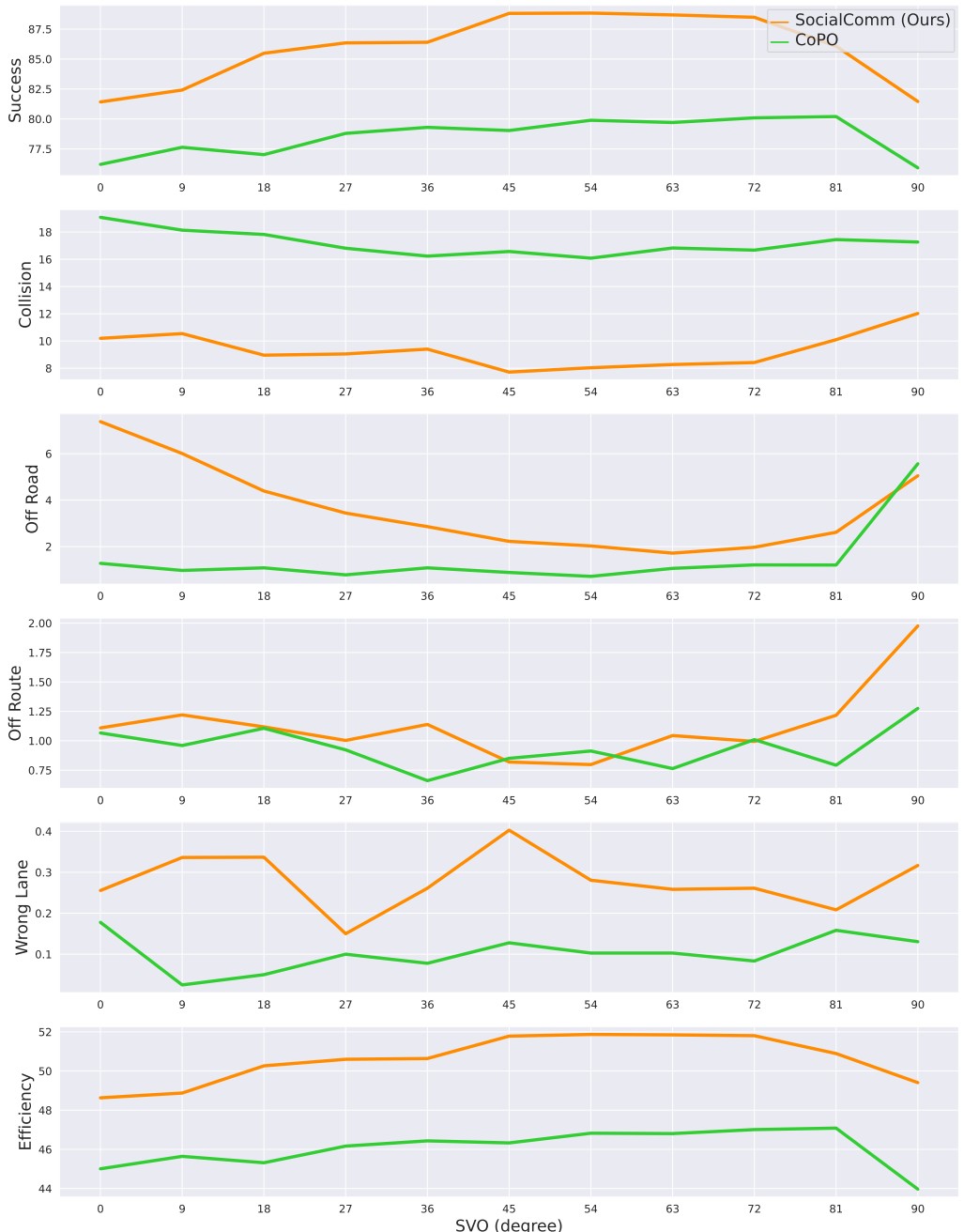

Figure 7: **The performance of CoPO and SocialComm (Ours) in `intersection`.**

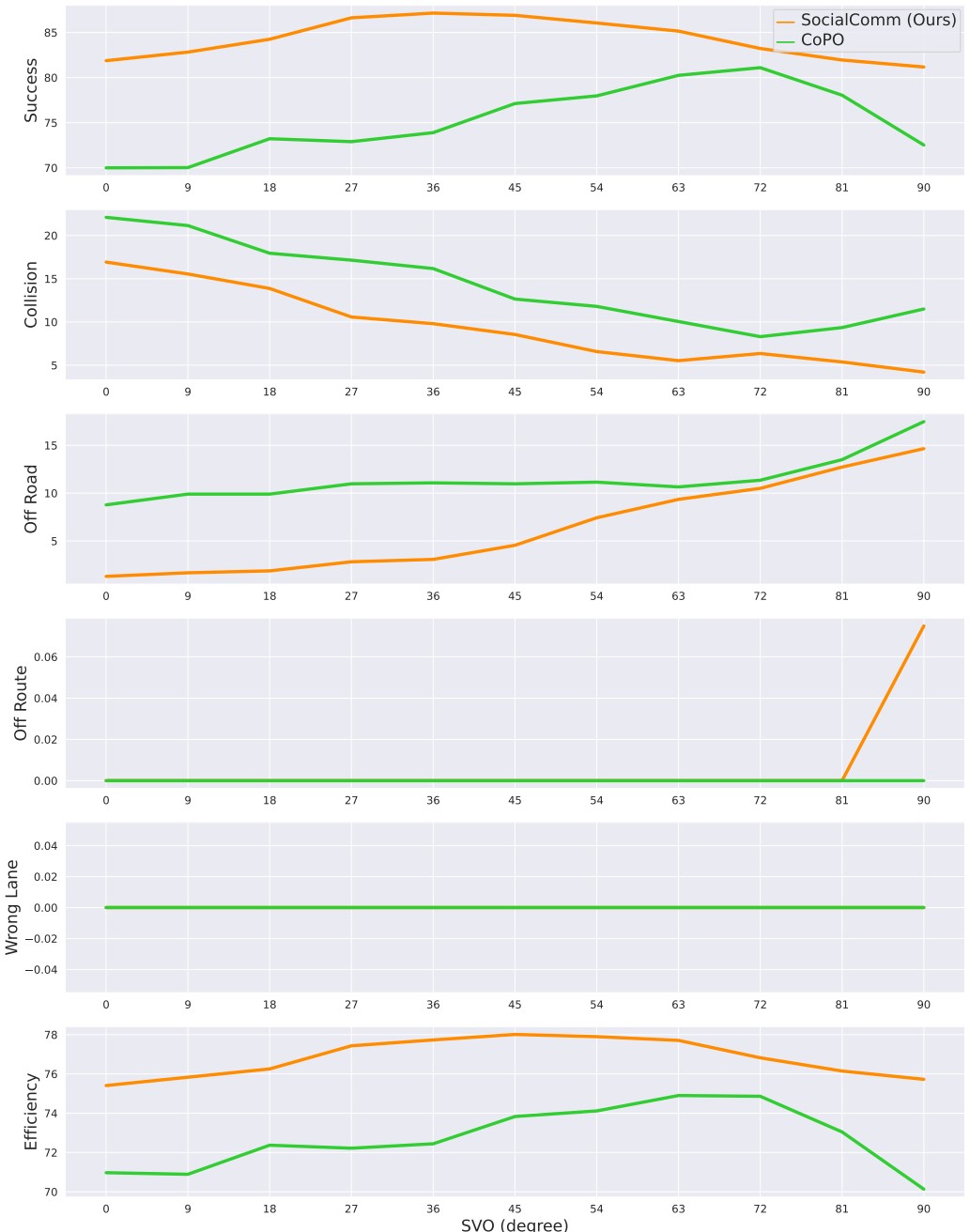

Figure 8: **The performance of CoPO and SocialComm (Ours) in `bottleneck`.**

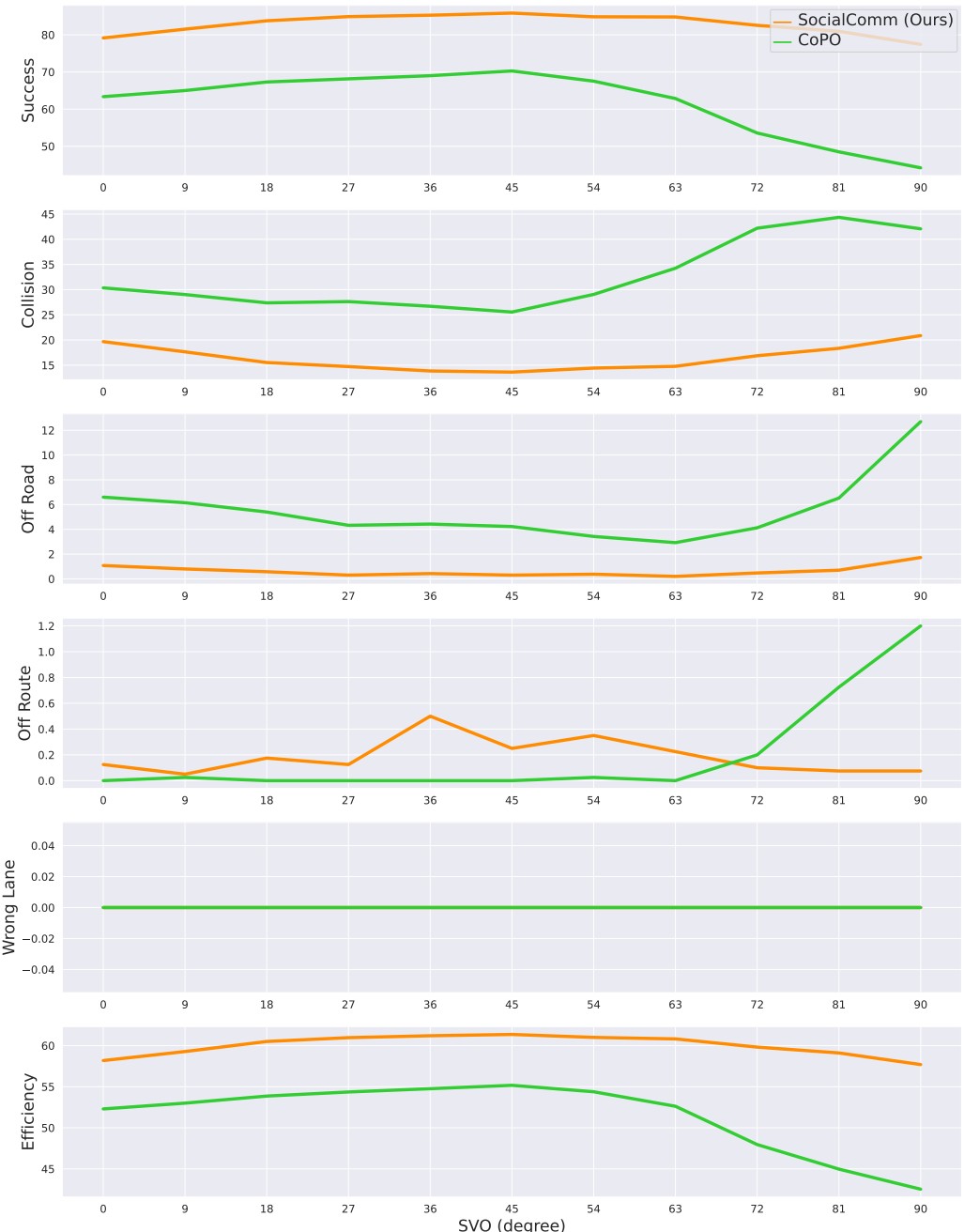

Figure 9: **The performance of CoPO and SocialComm (Ours) in `merge`.**

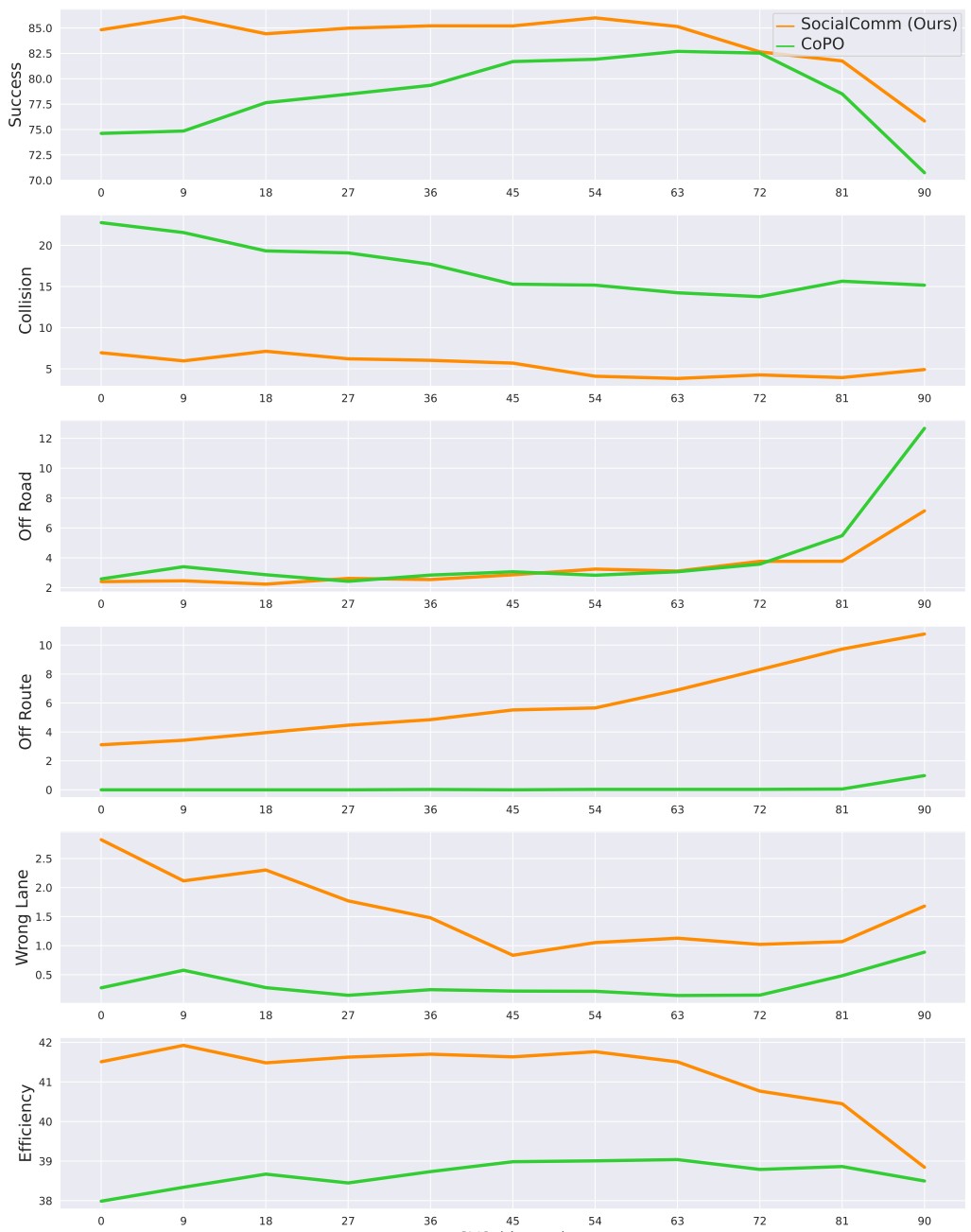

Figure 10: **The performance of CoPO and SocialComm (Ours) in `roundabout`.**

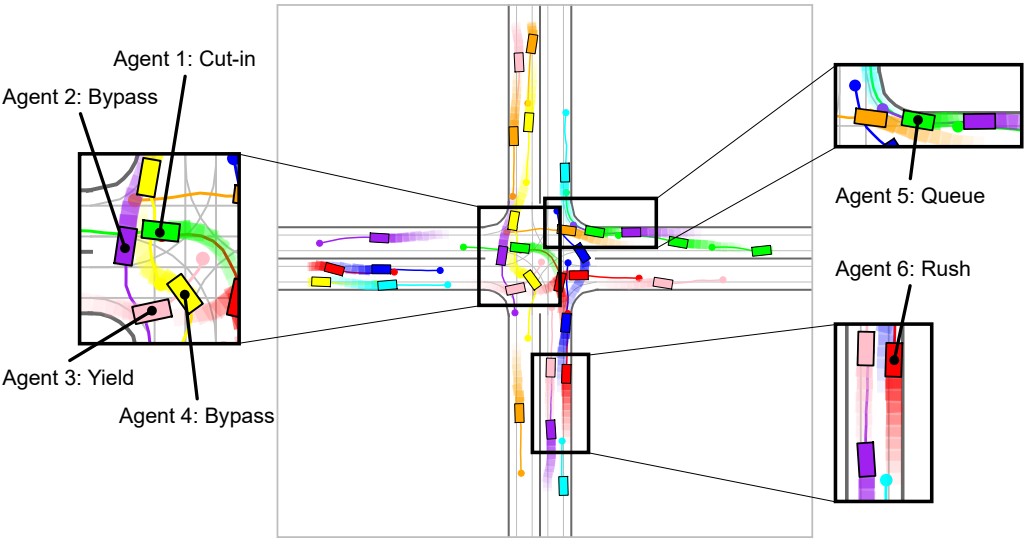

Figure 11: **Coordinated behaviors in `intersection`.**

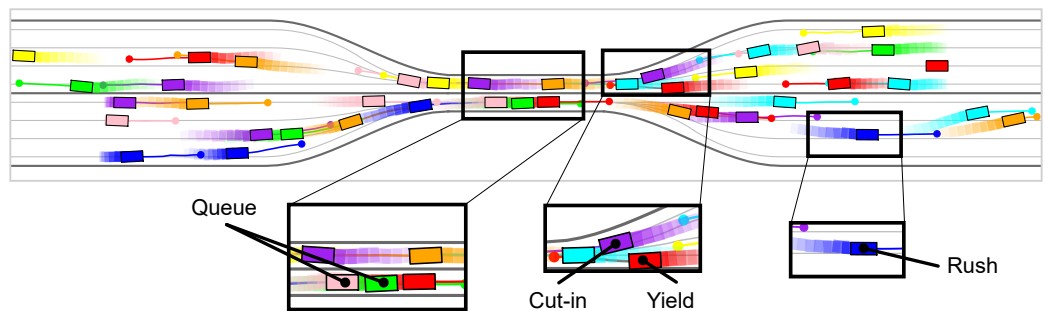

Figure 12: **Coordinated behaviors in `bottleneck`.**

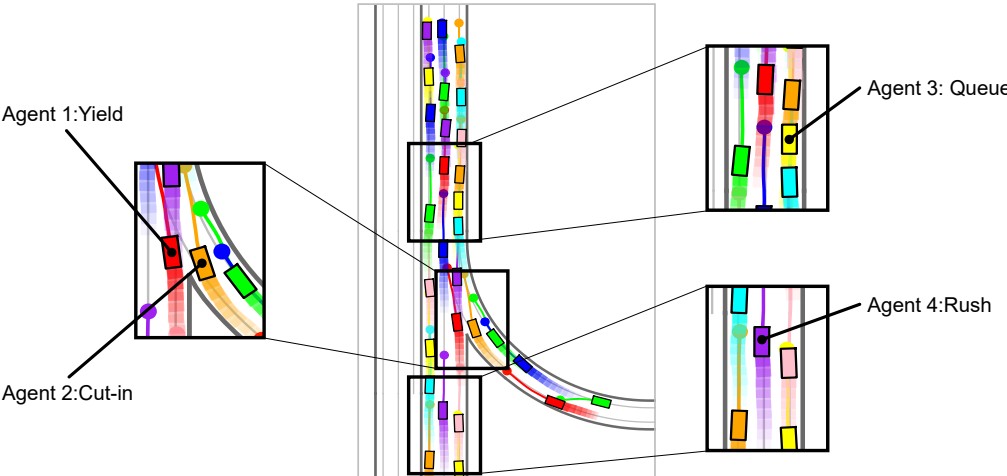

Figure 13: **Coordinated behaviors in `merge`.**

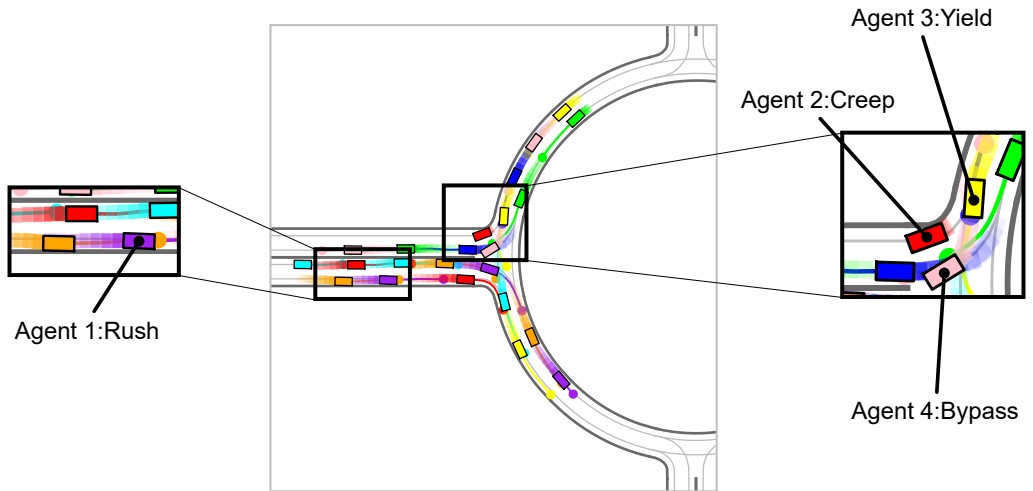

Figure 14: **Coordinated behaviors in `roundabout`.**

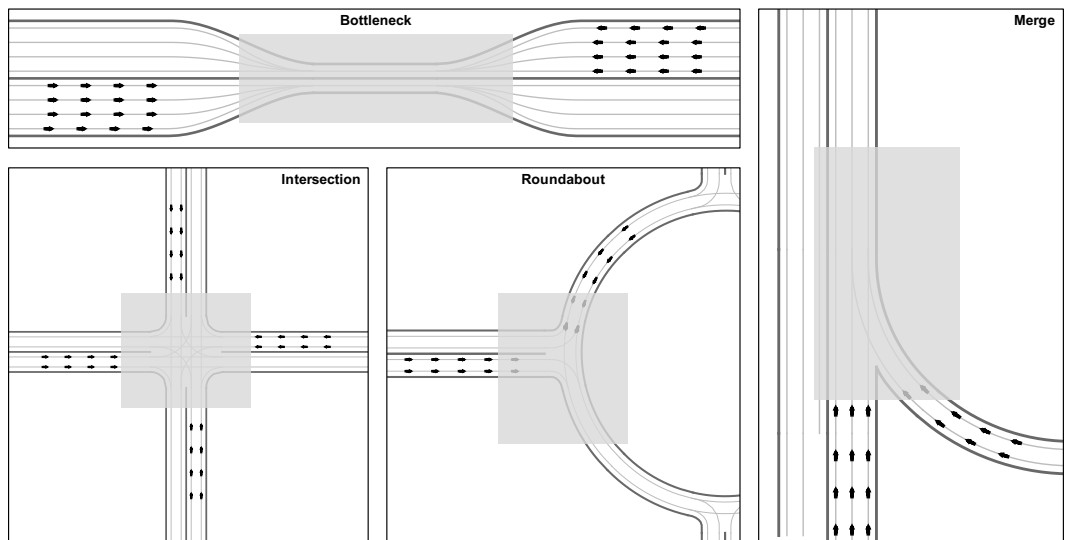

Figure 15: **The initial poses (makred as black arrows) and interaction zone (marked as gray rectangles) of each scenario.**

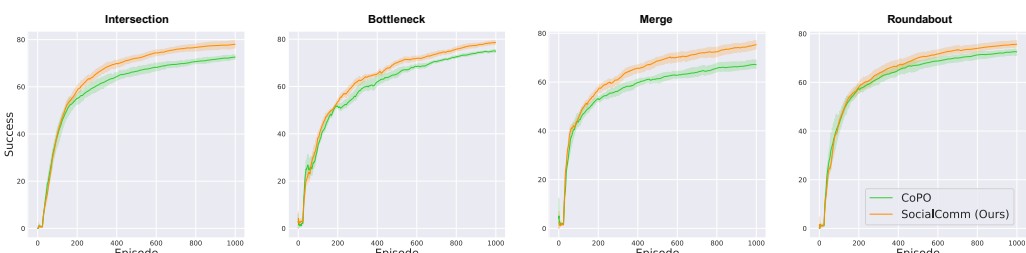

Figure 16: **Training success rates of CoPO and SocialComm (Ours).**

Table 5: **Zero-shot transfer performance in `intersection`.** Each subtable stores results of different driving policies in the same traffic flow. A "†" indicates our proposed method.

| Evaluate in IDM | Intersection | | | | | |
|---|---|---|---|---|---|---|
| | Success (↑) | Collision (↓) | Off Road (↓) | Off Route (↓) | Wrong Lane (↓) | Efficiency (↑) |
| VRL/IDM | 79.5 | 6.0 | 2.0 | 0.0 | 0.0 | 47.1 |
| VRL/FLOW | 67.0 | 30.5 | **0.5** | 0.5 | 0.5 | 39.5 |
| VRL/CoPO | 69.0 | 27.0 | **0.5** | **0.0** | **0.0** | **43.0** |
| RARL/FailMaker | 22.5 | **20.5** | 19.5 | 11.0 | **0.0** | 25.2 |
| VRL/SocialComm | 69.0 | 28.5 | 3.5 | 2.0 | **0.0** | 40.8 |
| M-RARL/SocialComm† | **71.5** | 21.5 | 6.0 | 1.0 | **0.0** | 42.1 |

| Evaluate in FLOW | Intersection | | | | | |
|---|---|---|---|---|---|---|
| | Success (↑) | Collision (↓) | Off Road (↓) | Off Route (↓) | Wrong Lane (↓) | Efficiency (↑) |
| VRL/IDM | 65.5 | 18.5 | 13.0 | **1.5** | 0.5 | 42.8 |
| VRL/FLOW | 87.5 | 6.5 | 1.5 | 0.0 | 0.0 | 51.5 |
| VRL/CoPO | 76.0 | 12.0 | **2.0** | 3.0 | **0.0** | 46.5 |
| RARL/FailMaker | 26.0 | 13.5 | 13.0 | 16.0 | 0.5 | 25.7 |
| VRL/SocialComm | 79.0 | 11.0 | **2.0** | 4.5 | **0.0** | 47.9 |
| M-RARL/SocialComm† | **83.5** | **8.0** | **2.0** | 3.0 | 0.5 | **49.7** |

| Evaluate in CoPO | Intersection | | | | | |
|---|---|---|---|---|---|---|
| | Success (↑) | Collision (↓) | Off Road (↓) | Off Route (↓) | Wrong Lane (↓) | Efficiency (↑) |
| VRL/IDM | 69.5 | 20.0 | 8.0 | 1.0 | 0.5 | 45.0 |
| VRL/FLOW | 75.5 | 20.0 | 1.5 | **0.0** | **0.0** | 46.1 |
| VRL/CoPO | 81.5 | 16.0 | 1.0 | 0.0 | 0.0 | 48.7 |
| RARL/FailMaker | 25.5 | **11.5** | 15.5 | 14.5 | 0.5 | 26.0 |
| VRL/SocialComm | 74.5 | 19.0 | 1.0 | 2.0 | **0.0** | 45.8 |
| M-RARL/SocialComm† | **79.5** | 16.5 | **0.0** | 1.0 | **0.0** | 47.7 |

| Evaluate in FailMaker | Intersection | | | | | |
|---|---|---|---|---|---|---|
| | Success (↑) | Collision (↓) | Off Road (↓) | Off Route (↓) | Wrong Lane (↓) | Efficiency (↑) |
| VRL/IDM | **55.0** | **22.5** | 22.0 | **0.0** | 0.5 | **37.3** |
| VRL/FLOW | 52.0 | 37.0 | 12.0 | **0.0** | **0.0** | 36.0 |
| VRL/CoPO | 51.0 | 43.5 | **4.0** | 1.0 | 0.5 | 34.7 |
| RARL/FailMaker | 26.5 | 11.5 | 17.5 | 14.5 | **0.0** | 25.9 |
| VRL/SocialComm | 51.5 | 33.0 | 16.5 | 1.5 | **0.0** | 36.5 |
| M-RARL/SocialComm† | 51.0 | 40.5 | 8.0 | **0.0** | **0.0** | 35.9 |

| Evaluate in SocialComm | Intersection | | | | | |
|---|---|---|---|---|---|---|
| | Success (↑) | Collision (↓) | Off Road (↓) | Off Route (↓) | Wrong Lane (↓) | Efficiency (↑) |
| VRL/IDM | 77.0 | **10.0** | 11.0 | 1.0 | **0.0** | 47.1 |
| VRL/FLOW | 84.0 | 13.5 | **1.0** | **0.5** | 0.5 | 50.1 |
| VRL/CoPO | 81.5 | 14.5 | **1.0** | 2.0 | **0.0** | 48.8 |
| RARL/FailMaker | 21.5 | 12.5 | 21.0 | 15.0 | 0.5 | 24.9 |
| VRL/SocialComm | 87.0 | 7.0 | 1.5 | 1.5 | 0.5 | 51.9 |
| M-RARL/SocialComm† | **85.5** | 11.0 | **1.0** | **0.5** | **0.0** | 50.9 |

| Evaluate in SocialComm (with Adv) | Intersection | | | | | |
|---|---|---|---|---|---|---|
| | Success (↑) | Collision (↓) | Off Road (↓) | Off Route (↓) | Wrong Lane (↓) | Efficiency (↑) |
| VRL/IDM | 74.5 | 14.5 | 7.5 | 2.5 | **0.0** | 45.9 |
| VRL/FLOW | 82.5 | 15.0 | 2.0 | **0.0** | **0.0** | 49.6 |
| VRL/CoPO | 78.5 | 18.5 | **1.5** | 1.0 | **0.0** | 47.5 |
| RARL/FailMaker | 24.0 | 12.5 | 15.5 | 18.0 | 0.5 | 25.3 |
| VRL/SocialComm | **85.0** | **9.0** | 3.0 | 1.0 | 0.5 | **50.8** |
| M-RARL/SocialComm† | 86.0 | 10.5 | 0.5 | 1.5 | 0.0 | 51.4 |

Table 6: **Zero-shot transfer performance in `bottleneck`.** Each subtable stores results of different driving policies in the same traffic flow. A "†" indicates our proposed method.

| Evaluate in IDM | Bottleneck | | | | | |
| --- | --- | --- | --- | --- | --- | --- |
| | Success (↑) | Collision (↓) | Off Road (↓) | Off Route (↓) | Wrong Lane (↓) | Efficiency (↑) |
| VRL/IDM | 91.5 | 6.0 | 2.5 | 0.0 | 0.0 | 72.8 |
| VRL/FLOW | 53.0 | 25.5 | 19.5 | 2.5 | **0.0** | 51.7 |
| VRL/CoPO | 64.5 | 24.0 | 10.5 | 2.5 | **0.0** | **59.6** |
| RARL/FailMaker | 50.0 | **10.0** | 37.5 | 3.0 | **0.0** | 33.9 |
| VRL/SocialComm | 67.0 | 20.5 | 12.5 | 0.0 | **0.0** | 55.8 |
| M-RARL/SocialComm† | **74.5** | 21.0 | **5.5** | 0.0 | **0.0** | 57.6 |

| Evaluate in FLOW | Bottleneck | | | | | |
| --- | --- | --- | --- | --- | --- | --- |
| | Success (↑) | Collision (↓) | Off Road (↓) | Off Route (↓) | Wrong Lane (↓) | Efficiency (↑) |
| VRL/IDM | 51.0 | 24.0 | 26.5 | 0.5 | **0.0** | 54.8 |
| VRL/FLOW | 79.0 | 18.5 | 2.0 | 0.5 | 0.0 | 76.1 |
| VRL/CoPO | 60.5 | 27.5 | 12.5 | 0.0 | **0.0** | 65.5 |
| RARL/FailMaker | 42.0 | **18.5** | 39.5 | 0.0 | **0.0** | 25.6 |
| VRL/SocialComm | 72.5 | 24.0 | 4.5 | 0.0 | **0.0** | **68.7** |
| M-RARL/SocialComm† | **73.0** | 25.5 | **2.0** | 0.0 | **0.0** | 68.4 |

| Evaluate in CoPO | Bottleneck | | | | | |
| --- | --- | --- | --- | --- | --- | --- |
| | Success (↑) | Collision (↓) | Off Road (↓) | Off Route (↓) | Wrong Lane (↓) | Efficiency (↑) |
| VRL/IDM | 48.5 | 16.5 | 33.0 | 2.0 | **0.0** | 57.2 |
| VRL/FLOW | 61.0 | **6.5** | 33.0 | 0.0 | **0.0** | 64.6 |
| VRL/CoPO | 89.0 | 9.0 | 1.5 | 0.5 | 0.0 | 81.8 |
| RARL/FailMaker | 51.0 | 11.0 | 34.5 | 3.5 | **0.0** | 35.1 |
| VRL/SocialComm | 65.0 | 30.5 | 5.5 | 0.0 | **0.0** | 67.2 |
| M-RARL/SocialComm† | **74.0** | 25.5 | **0.5** | 0.0 | **0.0** | **71.0** |

| Evaluate in FailMaker | Bottleneck | | | | | |
| --- | --- | --- | --- | --- | --- | --- |
| | Success (↑) | Collision (↓) | Off Road (↓) | Off Route (↓) | Wrong Lane (↓) | Efficiency (↑) |
| VRL/IDM | **20.5** | **59.0** | 26.0 | **0.0** | **0.0** | **34.5** |
| VRL/FLOW | 4.5 | 94.0 | 3.0 | **0.0** | **0.0** | 24.2 |
| VRL/CoPO | 6.5 | 88.5 | 6.5 | **0.0** | **0.0** | 27.5 |
| RARL/FailMaker | 38.0 | 23.0 | 38.5 | 0.5 | **0.0** | 26.1 |
| VRL/SocialComm | 2.5 | 87.5 | 13.5 | **0.0** | **0.0** | 24.3 |
| M-RARL/SocialComm† | 4.5 | 94.0 | **2.0** | **0.0** | **0.0** | 24.1 |

| Evaluate in SocialComm | Bottleneck | | | | | |
| --- | --- | --- | --- | --- | --- | --- |
| | Success (↑) | Collision (↓) | Off Road (↓) | Off Route (↓) | Wrong Lane (↓) | Efficiency (↑) |
| VRL/IDM | 52.5 | 26.0 | 21.0 | 0.5 | **0.0** | 58.3 |
| VRL/FLOW | 75.5 | 19.5 | 4.5 | 0.5 | **0.0** | 74.6 |
| VRL/CoPO | 71.5 | 13.5 | 15.5 | 1.0 | **0.0** | 70.9 |
| RARL/FailMaker | 36.0 | 23.5 | 42.0 | 0.5 | **0.0** | 26.6 |
| VRL/SocialComm | 91.0 | 4.0 | 5.0 | 0.0 | 0.0 | 81.3 |
| M-RARL/SocialComm† | **89.5** | **8.5** | **2.0** | 0.0 | **0.0** | **79.6** |

| Evaluate in SocialComm (with Adv) | Bottleneck | | | | | |
| --- | --- | --- | --- | --- | --- | --- |
| | Success (↑) | Collision (↓) | Off Road (↓) | Off Route (↓) | Wrong Lane (↓) | Efficiency (↑) |
| VRL/IDM | 50.5 | 27.0 | 22.0 | 0.5 | **0.0** | 57.0 |
| VRL/FLOW | 73.5 | 24.0 | **2.0** | 0.5 | **0.0** | 73.8 |
| VRL/CoPO | 69.5 | 11.0 | 19.0 | 1.0 | **0.0** | 69.9 |
| RARL/FailMaker | 36.5 | 19.0 | 43.5 | 1.0 | **0.0** | 27.7 |
| VRL/SocialComm | **85.0** | **7.0** | 8.0 | 0.0 | **0.0** | **78.3** |
| M-RARL/SocialComm† | 88.5 | 11.5 | 0.0 | 0.0 | 0.0 | 79.8 |

Table 7: **Zero-shot transfer performance in `merge`.** Each subtable stores results of different driving policies in the same traffic flow. A "†" indicates our proposed method.

| Evaluate in IDM | Merge | | | | | |
|---|---|---|---|---|---|---|
| | Success (↑) | Collision (↓) | Off Road (↓) | Off Route (↓) | Wrong Lane (↓) | Efficiency (↑) |
| VRL/IDM | 92.0 | 6.0 | 2.0 | 0.0 | 0.0 | 62.3 |
| VRL/FLOW | 41.5 | 55.5 | 7.5 | 0.5 | **0.0** | 29.1 |
| VRL/CoPO | 54.5 | 19.0 | 22.5 | 9.0 | **0.0** | 38.0 |
| RARL/FailMaker | 66.5 | **5.0** | 14.5 | 14.0 | **0.0** | 45.9 |
| VRL/SocialComm | 67.5 | 18.5 | 14.5 | 0.0 | 0.0 | 43.6 |
| M-RARL/SocialComm† | **85.0** | 15.0 | **0.0** | **0.0** | **0.0** | **53.2** |

| Evaluate in FLOW | Merge | | | | | |
|---|---|---|---|---|---|---|
| | Success (↑) | Collision (↓) | Off Road (↓) | Off Route (↓) | Wrong Lane (↓) | Efficiency (↑) |
| VRL/IDM | 35.0 | 38.0 | 27.5 | **0.0** | **0.0** | 40.5 |
| VRL/FLOW | 69.5 | 25.0 | 5.5 | 0.0 | 0.0 | 56.0 |
| VRL/CoPO | 52.0 | 35.0 | 12.0 | 1.0 | **0.0** | 48.2 |
| RARL/FailMaker | 17.0 | 40.5 | 33.5 | 9.5 | **0.0** | 24.2 |
| VRL/SocialComm | 62.5 | 33.5 | **4.0** | 0.0 | **0.0** | 54.5 |
| M-RARL/SocialComm† | **67.5** | **21.5** | 11.0 | 0.0 | **0.0** | **56.0** |

| Evaluate in CoPO | Merge | | | | | |
|---|---|---|---|---|---|---|
| | Success (↑) | Collision (↓) | Off Road (↓) | Off Route (↓) | Wrong Lane (↓) | Efficiency (↑) |
| VRL/IDM | 36.5 | 44.5 | 19.0 | **0.0** | **0.0** | 40.0 |
| VRL/FLOW | 50.0 | 44.0 | 5.5 | 0.5 | **0.0** | 46.6 |
| VRL/CoPO | 70.5 | 24.5 | 5.0 | 0.0 | 0.0 | 55.4 |
| RARL/FailMaker | 17.0 | 39.0 | 34.0 | 10.0 | **0.0** | 23.5 |
| VRL/SocialComm | 57.0 | **37.5** | 5.5 | 0.0 | **0.0** | 49.9 |
| M-RARL/SocialComm† | **59.5** | 40.0 | **0.5** | 0.0 | **0.0** | **52.0** |

| Evaluate in FailMaker | Merge | | | | | |
|---|---|---|---|---|---|---|
| | Success (↑) | Collision (↓) | Off Road (↓) | Off Route (↓) | Wrong Lane (↓) | Efficiency (↑) |
| VRL/IDM | 9.0 | **74.5** | 18.0 | **0.0** | **0.0** | 15.1 |
| VRL/FLOW | 9.0 | 81.0 | 11.5 | **0.0** | **0.0** | 17.5 |
| VRL/CoPO | 10.0 | 83.5 | 10.0 | **0.0** | **0.0** | **19.2** |
| RARL/FailMaker | 19.5 | 37.5 | 39.5 | 6.5 | 0.0 | 21.6 |
| VRL/SocialComm | 8.5 | 88.0 | **4.0** | **0.0** | **0.0** | 16.8 |
| M-RARL/SocialComm† | **12.0** | 78.5 | 11.0 | **0.0** | **0.0** | 18.5 |

| Evaluate in SocialComm | Merge | | | | | |
|---|---|---|---|---|---|---|
| | Success (↑) | Collision (↓) | Off Road (↓) | Off Route (↓) | Wrong Lane (↓) | Efficiency (↑) |
| VRL/IDM | 65.0 | 16.0 | 19.0 | **0.0** | **0.0** | 51.3 |
| VRL/FLOW | 71.5 | 21.0 | 7.5 | **0.0** | **0.0** | 55.0 |
| VRL/CoPO | 70.0 | 24.5 | 5.5 | 0.5 | **0.0** | 55.1 |
| RARL/FailMaker | 18.5 | 29.0 | 47.5 | 6.5 | **0.0** | 27.6 |
| VRL/SocialComm | 84.5 | 14.0 | 1.5 | 0.0 | 0.0 | 60.7 |
| M-RARL/SocialComm† | **83.5** | **12.5** | **3.5** | 0.5 | **0.0** | **61.2** |

| Evaluate in SocialComm (with Adv) | Merge | | | | | |
|---|---|---|---|---|---|---|
| | Success (↑) | Collision (↓) | Off Road (↓) | Off Route (↓) | Wrong Lane (↓) | Efficiency (↑) |
| VRL/IDM | 64.5 | 18.0 | 17.5 | **0.0** | **0.0** | 51.1 |
| VRL/FLOW | 69.5 | 22.5 | 8.0 | **0.0** | **0.0** | 54.1 |
| VRL/CoPO | 66.0 | 23.5 | 10.5 | 0.5 | **0.0** | 53.4 |
| RARL/FailMaker | 18.5 | 25.0 | 50.5 | 7.0 | **0.0** | 27.2 |
| VRL/SocialComm | **83.5** | **15.0** | 1.5 | 0.0 | **0.0** | **61.0** |
| M-RARL/SocialComm† | 85.0 | 13.5 | 1.5 | 0.0 | 0.0 | 61.3 |

Table 8: **Zero-shot transfer performance in `roundabout`.** Each subtable stores results of different driving policies in the same traffic flow. A "†" indicates our proposed method.

| Evaluate in IDM | Roundabout | | | | | |
|---|---|---|---|---|---|---|
| | Success (↑) | Collision (↓) | Off Road (↓) | Off Route (↓) | Wrong Lane (↓) | Efficiency (↑) |
| VRL/IDM | 83.0 | 9.0 | 5.5 | 0.0 | 2.5 | 38.7 |
| VRL/FLOW | 44.5 | 51.5 | **4.0** | 1.0 | **0.0** | 23.8 |
| VRL/CoPO | 59.0 | 33.0 | 9.5 | **0.0** | **0.0** | 31.5 |
| RARL/FailMaker | 32.5 | **16.0** | 55.5 | 6.0 | 0.5 | 19.7 |
| VRL/SocialComm | 40.5 | 30.5 | 31.0 | 0.5 | 1.0 | 21.1 |
| M-RARL/SocialComm† | **63.0** | 18.5 | 17.0 | **0.0** | 1.5 | **32.3** |

| Evaluate in FLOW | Roundabout | | | | | |
|---|---|---|---|---|---|---|
| | Success (↑) | Collision (↓) | Off Road (↓) | Off Route (↓) | Wrong Lane (↓) | Efficiency (↑) |
| VRL/IDM | 54.0 | 27.0 | 18.5 | **0.0** | 0.5 | 32.7 |
| VRL/FLOW | 84.0 | 11.5 | 4.5 | 0.0 | 0.0 | 42.7 |
| VRL/CoPO | 61.5 | 29.5 | 9.0 | **0.0** | **0.0** | 38.4 |
| RARL/FailMaker | 30.5 | **24.0** | 46.0 | 5.0 | 0.5 | 17.7 |
| VRL/SocialComm | 66.5 | 25.5 | 9.5 | **0.0** | **0.0** | 38.6 |
| M-RARL/SocialComm† | **70.5** | 26.5 | **2.5** | **0.0** | 0.5 | **40.1** |

| Evaluate in CoPO | Roundabout | | | | | |
|---|---|---|---|---|---|---|
| | Success (↑) | Collision (↓) | Off Road (↓) | Off Route (↓) | Wrong Lane (↓) | Efficiency (↑) |
| VRL/IDM | 55.5 | 18.5 | 23.0 | 0.5 | 2.5 | 32.4 |
| VRL/FLOW | 62.0 | **12.0** | 24.0 | 0.0 | 2.0 | 35.7 |
| VRL/CoPO | 86.5 | 12.0 | 1.5 | 0.0 | 0.0 | 41.8 |
| RARL/FailMaker | 32.0 | 25.0 | 46.5 | 4.5 | **0.5** | 17.2 |
| VRL/SocialComm | 70.0 | 26.5 | **3.5** | **0.0** | 0.5 | 38.8 |
| M-RARL/SocialComm† | **76.5** | 17.5 | 5.5 | **0.0** | 0.5 | **39.9** |

| Evaluate in FailMaker | Roundabout | | | | | |
|---|---|---|---|---|---|---|
| | Success (↑) | Collision (↓) | Off Road (↓) | Off Route (↓) | Wrong Lane (↓) | Efficiency (↑) |
| VRL/IDM | 19.5 | 64.0 | 20.5 | **0.0** | 0.5 | 20.2 |
| VRL/FLOW | **29.5** | **57.5** | 14.0 | **0.0** | **0.0** | **25.0** |
| VRL/CoPO | 21.0 | 70.5 | **10.0** | **0.0** | 0.5 | 22.7 |
| RARL/FailMaker | 29.5 | 17.0 | 51.5 | 5.0 | **0.0** | 14.5 |
| VRL/SocialComm | 17.0 | 63.0 | 24.5 | **0.0** | **0.0** | 19.9 |
| M-RARL/SocialComm† | 18.5 | **57.5** | 25.5 | **0.0** | **0.0** | 20.5 |

| Evaluate in SocialComm | Roundabout | | | | | |
|---|---|---|---|---|---|---|
| | Success (↑) | Collision (↓) | Off Road (↓) | Off Route (↓) | Wrong Lane (↓) | Efficiency (↑) |
| VRL/IDM | 75.5 | **9.5** | 13.0 | **0.0** | 2.0 | 38.0 |
| VRL/FLOW | 74.0 | 12.0 | 14.0 | **0.0** | **0.0** | 39.7 |
| VRL/CoPO | 82.0 | 13.5 | **1.5** | 1.5 | 2.5 | 42.4 |
| RARL/FailMaker | 30.0 | 16.5 | 54.5 | 5.0 | **0.0** | 17.7 |
| VRL/SocialComm | 87.5 | 9.0 | 3.5 | **0.0** | **0.0** | 43.2 |
| M-RARL/SocialComm† | **86.0** | 12.0 | 2.0 | **0.0** | **0.0** | **42.8** |

| Evaluate in SocialComm (with Adv) | Roundabout | | | | | |
|---|---|---|---|---|---|---|
| | Success (↑) | Collision (↓) | Off Road (↓) | Off Route (↓) | Wrong Lane (↓) | Efficiency (↑) |
| VRL/IDM | 72.5 | 13.0 | 13.0 | **0.0** | 1.5 | 37.8 |
| VRL/FLOW | 81.0 | **4.0** | 14.5 | 0.5 | **0.0** | 41.2 |
| VRL/CoPO | 80.5 | 17.5 | **1.5** | 1.0 | **0.0** | 42.2 |
| RARL/FailMaker | 31.0 | 15.5 | 56.0 | 4.5 | 0.5 | 18.2 |
| VRL/SocialComm | **84.5** | 11.5 | 5.0 | **0.0** | **0.0** | **42.3** |
| M-RARL/SocialComm† | 87.0 | 10.5 | 2.0 | 0.0 | 0.5 | 42.9 |

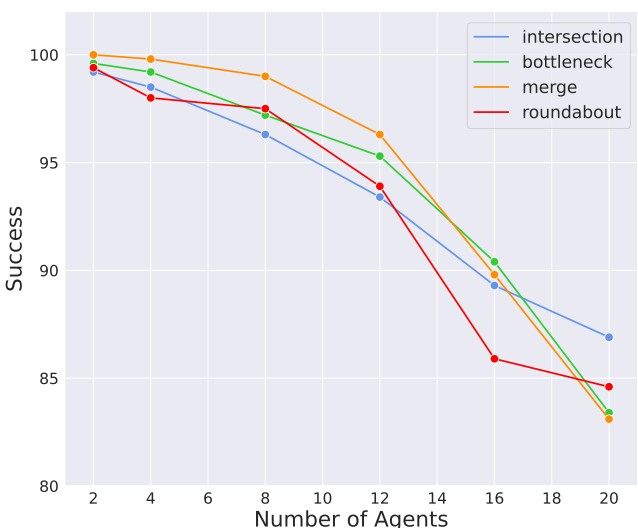

Figure 17: **Success rates with different number of agents.**

