# OpenReview forum: "Adversarial Driving Policy Learning by Misunderstanding the Traffic Flow"
_ICLR.cc/2023/Conference — Submitted to ICLR 2023_

### Official Review · Reviewer_MJTo · 2022-10-15

**Confidence:** 3
**Correctness:** 3
**Technical Novelty And Significance:** 3
**Empirical Novelty And Significance:** 2
**Recommendation:** 5

**Clarity, Quality, Novelty And Reproducibility:**

## Clarity

Some parts are not very clear. For example, the usage of SVOs and the key difference with [Peng 2021] need to be further clarified and highlighted. Experiment setting details are also missing.


1.Can you further clarify the key difference (using the notations in Sec 3.1), between the current work and [Peng 2021] in terms of “agents in their traffic flow have no access to other agents’ SVOs, leading to conservative behaviors”? Is it basically the policy beta will only have O_i and C_i as the inputs but not the C_j for others? Are all the C_i for each agent i also learned during the training?

2.How many different settings were used for each scenario during the evaluation?

3.in (2), why the optimization for beta is to solve n separate optimization objectives rather than their sum?

## Quality

The quantitative and qualitative results are moderately decent. One drawback is the realism issue I mentioned earlier. Scalability is also suspicious given it only has very limited scenario settings. However, this is somewhat acceptable given previous works e.g., [Peng 2021] along this line also does not evaluate realism and has limited scenarios to show effectiveness.

## Reproducibility

It is hard to reproduce given code is not available, details of simulator and experiment settings are all missing.


## Minor Issues:

-Sec 3.2 - state space - last two lines, probably should not overload notation c_i here.


**Strength And Weaknesses:**

## Pros:

1.Leveraging SVOs for adversarial attack are moderately novel and its effect on improving zero-shot transfer of the driving policy seems promising.

2.generalization of the training policy is an important problem

## Cons:

1.Communicating SVOs among agents seem to be a relatively incremental improvement over [Peng 2021].

2.One issue is that the realism evaluation of the settings is missing
Are the generated trajectories realistic and initial positions realistic? One way to evaluate if the trajectories are realistic is to look into acceleration (both longitudinal and latitudinal) and jerk distributions and compare them against some similar scenarios in real world datasets (e.g., nuScenes and Waymo Open).

3.some details are missing
see “Clarity, Quality, Novelty And Reproducibility” below.

4.effect of adversary is not that large
Table 1 success rates are not influenced much percentage wise. One usually expects the attack to have a stronger effect.


**Summary Of The Paper:**

This paper proposes a robust RL policy training framework which applies adversarial training using coordinated traffic flow. For building the traffic flow, it uses the so-called “Social Value Orientations (SVOs)” (which is a weight that balances the proportion of each agent’s own driving reward and its surrounding agents driving rewards in the agent’s aggregated optimization reward) and allows the communication of SVOs among the background agents in the traffic flow. Adversary emerges when the SVOs of the driving agent sent to the traffic flow agents are perturbed. The problem can then be turned into a min-max optimization problem where the driving policy maximizes its reward and a spurious adversarial policy minimizes it through changing the SVOs seen by the agents. The proposed method is evaluated in terms of the safety, speed, and success rate of the generated traffic flows, the effect of adversary based on coordinated traffic flow, and the zero-shot transfer of the driving policy.

**Summary Of The Review:**

Overall the approach is moderately interesting and the results seem to be promising. However, some clarification needs to be made regarding the novelty of the method and the details of evaluation for a better evaluation of the paper.

---

> ### Author Response · Authors · 2022-11-16
> **Response to Reviewer MJTo (1-2)**
>
> **Q5: How many different settings were used for each scenario during the evaluation?**
>
> For evaluating various methods fairly, we offline generate 200 cases (with different global paths, initial poses, and SVOs) for each scenario. One case can be regarded as one setting. Please refer to Q2 in the common response for more details.
>
> ---
>
> **Q6: in (2), why the optimization for beta is to solve n separate optimization objectives rather than their sum?**
>
> You are right. Since we use parameter sharing technique, optimization over their sum makes more sense. We will correct this issue in the next version.
>
> ---
>
> **Q7: Scalability is also suspicious given it only has very limited scenario settings.**
>
> Scalability is meaningful since real-world driving contains numerous scenarios. In our paper, we pick up four representative highly interactive scenarios where plenty of agents need to continually interact with each other while prior works mainly focus on one or two scenarios with only a few agents. Besides, we find it hard for one single neural network to achieve high performance in all scenarios. In the future, we aim to propose a training pipeline to obtain policies that can achieve this.
>
> ---
>
> **Q8: It is hard to reproduce given code is not available, details of simulator and experiment settings are all missing.**
>
> We will open-source our simulator in one month. Our training code is based on https://github.com/pranz24/pytorch-soft-actor-critic. Please refer to the common response for more details about our simulator and experiment settings.
>
> ---
>
> **Q9: Sec 3.2 - state space - last two lines, probably should not overload notation c_i here.**
>
> Thank you for pointing out this issue. We will select another notation for lane width in the next version.
>
> ---
>
> **References:**
>
> [1] Tsun-Hsuan Wang, Sivabalan Manivasagam, Ming Liang, Bin Yang, Wenyuan Zeng, and Raquel Urtasun. V2vnet: Vehicle-to-vehicle communication for joint perception and prediction. In European Conference on Computer Vision, pp. 605–621. Springer, 2020.
>
> [2] Yiming Li, Shunli Ren, Pengxiang Wu, Siheng Chen, Chen Feng, and Wenjun Zhang. Learning distilled collaboration graph for multi-agent perception. Advances in Neural Information Processing Systems, 34:29541–29552, 2021.
>
> [3] Yiming Li, Dekun Ma, Ziyan An, Zixun Wang, Yiqi Zhong, Siheng Chen, and Chen Feng. V2x- sim: Multi-agent collaborative perception dataset and benchmark for autonomous driving. IEEE Robotics and Automation Letters, 7(4):10914–10921, 2022.
>
> [4] Jiaxun Cui, Hang Qiu, Dian Chen, Peter Stone, and Yuke Zhu. Coopernaut: End-to-end driving with cooperative perception for networked vehicles. In Proceedings of the IEEE/CVF Conference on Computer Vision and Pattern Recognition, pp. 17252–17262, 2022.
>
> [5] Simon Suo, Sebastian Regalado, Sergio Casas, and Raquel Urtasun. Trafficsim: Learning to simulate realistic multi-agent behaviors. In Proceedings of the IEEE/CVF Conference on Computer Vision and Pattern Recognition, pp. 10400–10409, 2021.
>
> [6] Luca Bergamini, Yawei Ye, Oliver Scheel, Long Chen, Chih Hu, Luca Del Pero, Błaz ̇ej Osin ́ski, Hugo Grimmett, and Peter Ondruska. Simnet: Learning reactive self-driving simulations from real-world observations. In 2021 IEEE International Conference on Robotics and Automation (ICRA), pp. 5119–5125. IEEE, 2021.

---

> ### Author Response · Authors · 2022-11-16
> **Response to Reviewer MJTo (1-1)**
>
> Thank you very much for your valuable feedback. We appreciate that you find our paper novel and promising. We address additional comments below.
>
> ---
>
> **Q1: Communicating SVOs among agents seem to be a relatively incremental improvement over [Peng 2021].**
>
> Communication in multi-agent systems is an active area where agents can make up for their invisible information with the help of some auxiliary information in a dynamic environment, so as to learn their own policies efficiently. What, how, and when to communicate are seminal problems. For autonomous driving, existing works mainly focus on investigating what kind of information to communicate can benefit perception and prediction systems [1,2,3,4], while seldom researches focus on the effect of communication in decision-making, especially in dense traffic flows. In our paper, we demonstrate that communicating SVOs improves coordination level, which answers the question "what to communicate".
>
> Besides, we cannot straightforwardly follow settings in [Peng 2021] to communicate SVOs by utilizing Lidar-like observation. In [Peng 2021], Lidar-like observation is a vector of 72 occupancy measures (5 degrees per measure) of the nearby environments. The information about a specific surrounding agent is missing. However, agent-specific information is indispensable when communicating SVOs, since agents need to be aware of their communication participants. Based on this, we utilize vectorized representation which contains agent-specific information and allows communication with each other.
>
> ---
>
> **Q2: One issue is that the realism evaluation of the settings is missing. Are the generated trajectories realistic and initial positions realistic?**
>
> We appreciate how carefully you have thought about this problem. Compared to real-world datasets such as nuScenes and Waymo Open, the generated trajectories and initial positions are less realistic till now. The reasons are two-fold. First, we have not taken realism metrics into consideration of our reward function, which is still an open problem. Second, our scenarios are more complex and are not covered in real-world datasets. For instance, $\texttt{intersection}$ allows agents to pass through the intersection from all directions simultaneously while most real-world intersections rely on traffic lights to moderate agents.
>
> Importantly, realism in dense traffic flow is a challenging and unsolved problem. Existing works investigate this problem with multi-agent imitation learning (from trajectory prediction community) [5,6]. However, IL techniques suffer from distribution shift and causal confusion, leading to poor closed-loop performance / reactivity. MARL shows strong reactivity, but designing reward functions that make agents realistic is challenging. We believe that a proper combination of IL and RL is a promising path to tackle this problem and will be a crucial part of our future work.
>
> ---
>
> **Q3: effect of adversary is not that large Table 1 success rates are not influenced much percentage wise. One usually expects the attack to have a stronger effect.**
>
> Your sense is right. Existing adversarial attacks do have a strong effect on metrics of driving policy. However, as described in Sec 1, this framework is impractical in dense traffic flows. Since it is almost impossible for the driving policy to resist such strong disturbances, leading to terrible transferability of driving policy, as shown in Figure 6, Table 5-8 in appendix. Therefore, we propose a "weaker" adversarial strategy that is capable of improving driving policy's robustness to unseen environments in dense traffic flows. Our adversary is "weaker" since background agents in our traffic flow aim to coordinate rather than attack with each other. As shown in Table 5-8 in appendix, FailMaker makes collision rates higher compared to other traffic flows.
>
> Last but not the least, our main results demonstrate that "weaker" adversary improves the transferability of driving policy to unseen traffic flows. The key is to construct a traffic flow where background agents coordinate with each other and still generate adversarial behaviors towards the driving agent.
>
> ---
>
> **Q4: Can you further clarify the key difference (using the notations in Sec 3.1), between the current work and [Peng 2021] in terms of “agents in their traffic flow have no access to other agents’ SVOs, leading to conservative behaviors”? Is it basically the policy beta will only have O_i and C_i as the inputs but not the C_j for others? Are all the C_i for each agent i also learned during the training?**
>
> You are right. For a fair comparison, the difference between our traffic flow and CoPO we implement is that $\beta$ in our traffic flow inputs $c_j$ while CoPO does not. In our traffic flow, each agent's SVO $c_i$ is sampled uniformly from $[0, \frac{\pi}{2}]$ and kept fixed in each episode during training.

---

> ### Author Response · Authors · 2022-12-08
> **Looking forward to further discussions**
>
> Dear Reviewer,
>
> We hope that our response can address your concerns. As the deadline for the discussion period is approaching, we would appreciate it if you could let us know whether there are any further questions about the paper or the response. We are looking forward to further discussions.
>
> Best wishes, Authors

---

### Official Review · Reviewer_o4m3 · 2022-10-25

**Confidence:** 3
**Correctness:** 3
**Technical Novelty And Significance:** 3
**Empirical Novelty And Significance:** 3
**Recommendation:** 5

**Clarity, Quality, Novelty And Reproducibility:**

**Clarity, Quality:** overall good quality and clarity if some of the figures/tables are edited.
**Novelty:** their method for adversarially attacking SVO seems relatively novel.
**Reproducibility:** the authors provide details about their architecture and learning algorithm, but I am unsure if this is reproducible without more training details and especially the simulator.

**Strength And Weaknesses:**

### Strengths

* Experiments show clear improvements over IDM, FLOW, COPO
* Overall writing is good, mathematical notion is for the most part clear

### Weaknesses

* Reproducibility - all experiments were performed on "our internal driving simulator". Can the authors give more details on this? This is a major factor in my decision.
* The experiments were all done in this one simulator and it is unclear if the effectiveness of this strategy translates to other domains.
* Writing/figure quality (more details below) - a couple figure/tables are quite hard to understand and could be presented better.

### Minor Issues / Questions

* What is the mathematical equation for the reward function? Were any other reward functions tried?
* Algorithm 1: in my opinion this is described well by Sec 4.2 and can be taken out.
* Table 1: Highlighted colors seem arbitrary - why is the "0.0" colored in "Ours + Wrong Lane"?
* Figure 6 took quite a while to understand what I was looking at. FailMaker seems bad and can be taken out and just described in text.

**Summary Of The Paper:**

This work builds a framework for traffic simulation and propose a form of adversarial training to improve driving policies.
In their traffic flow, they assume the agents allowed to communicate *social value orientations* (SVOs) and introduce an adversarial agent that tries to minimize the overall reward for the driving agent by only modifying the SVO.
They show strong improvements on their coordinated traffic flow task on a variety of subtasks (intersection, bottleneck, merge, roundabout).

**Summary Of The Review:**

This work introduces a fairly novel approach and overall shows improved performance across various scenarios in their simulator. The major factor in my current decision is on reproducibility.

---

> ### Author Response · Authors · 2022-11-16
> **Response to Reviewer o4m3 (1-3)**
>
>
> **Q4: Table 1: Highlighted colors seem arbitrary - why is the "0.0" colored in "Ours + Wrong Lane"?**
>
> This is our mistake. Thank you very much for pointing it out! We will correct this mistake in the next version.
>
> ---
>
> **Q5: Figure 6 took quite a while to understand what I was looking at. FailMaker seems bad and can be taken out and just described in text.**
>
> Thank you for pointing out this issue. We apologize for the confusion we introduce and will take your advice in the next version of this paper.
>
> ---
>
> **Q6: I am unsure if this is reproducible without more training details and especially the simulator.**
>
> We will open-source our simulator in one month. Our training code is based on https://github.com/pranz24/pytorch-soft-actor-critic. Please refer to the Q1 in common response for more details about our simulator and Q2 in the common response for more training details.
>
> ---
>
> **References:**
>
> [1] Alexey Dosovitskiy, German Ros, Felipe Codevilla, Antonio Lopez, and Vladlen Koltun. Carla: An open urban driving simulator. In Conference on robot learning, pp. 1–16. PMLR, 2017.
>
> [2] W Bradley Knox, Alessandro Allievi, Holger Banzhaf, Felix Schmitt, and Peter Stone. Reward (mis) design for autonomous driving. arXiv preprint arXiv:2104.13906, 2021.
>
> [3] Jingke Wang, Yue Wang, Dongkun Zhang, Yezhou Yang, and Rong Xiong. Learning hierarchical behavior and motion planning for autonomous driving. In 2020 IEEE/RSJ International Conference on Intelligent Robots and Systems (IROS), pp. 2235–2242. IEEE, 2020.
>
> [4] Xiaodan Liang, Tairui Wang, Luona Yang, and Eric Xing. Cirl: Controllable imitative reinforcement learning for vision-based self-driving. In Proceedings of the European conference on computer vision (ECCV), pp. 584–599, 2018.
>
> [5] Marin Toromanoff, Emilie Wirbel, and Fabien Moutarde. End-to-end model-free reinforcement learning for urban driving using implicit affordances. In Proceedings of the IEEE/CVF conference on computer vision and pattern recognition, pp. 7153–7162, 2020.

---

> ### Author Response · Authors · 2022-11-16
> **Response to Reviewer o4m3 (1-2)**
>
> **Q3: What is the mathematical equation for the reward function? Were any other reward functions tried?**
>
> As mentioned in Sec 3.2, we design a near-sparse reward function containing a dense reward ($R_{speed}$) for stimulating driving fast and a sparse reward ($R_{fail}$) for penalizing catastrophic failures. Catastrophic failures include collision with other agents, deviation from drivable area, driving too far from global path, and crashing into wrong lane.
>
> The dense reward $R_{speed}$ is defined as follow:
> $$
> R_{speed} = 2 \frac{v}{v_{max}} -1, \quad v \in [0, v_{max}]
> $$
> where $v$ and $v_{max}$ is the agent's current and maximum speed respectively. Since $v \in [0, v_{max}]$, $R_{speed} \in [-1,1]$ is bounded. Intuitively, this reward term encourages high driving efficiency and penalizes low driving speed.
>
> The sparse reward $R_{fail}$ is defined as follow:
> $$
> R_{fail} = - \mathbf{1}(\text{Collision} \ \lor \ \text{Off Road} \ \lor \ \text{Off Route} \ \lor \ \text{Wrong Lane})
> $$
> where $\mathbf{1}$ is the indicator function.
>
> The overall reward function is the combination of above items:
> $$
> R = \alpha R_{speed} + \beta R_{fail}
> $$
> In our implementation, $\alpha = 0.07$ and $\beta = 5$. Note that $\beta$ is about 100 times bigger than $\alpha$, this is because the number of steps is up to 100 in each episode.
>
> Actually, we have tried two different reward settings.
>
> One is to vary $\alpha$. We find that training with smaller $\alpha$ (like $0.03$ or $0.04$), the agents refuse to move forward and brake from beginning to end. This is probably due to that the failure penalty $R_{fail}$ is too strong for agents to capture the effect of $R_{speed}$. Under this circumstance, agents choose the safest action, i.e., stopping, to avoid large penalty. We hence increase $\alpha$ to $0.07$ and find it works well.
>
> Another trial is to remove the dense reward term $R_{speed}$ and add a positive sparse reward that stimulates goal-reaching. In this setting, the reward function $R$ is purely sparse, which is a well-known challenge to optimize with in RL community. Utilizing such sparse reward, we find agents also choose to stop all along. The reason is that in the early stage of training, agents always fail and have no access to positive incentives. The highest reward one agent can receive is 0, which represents that the agent stops to avoid receiving negative reward. Once stuck into this local optimum, there is no way to escape. Instead of designing delicate curriculum learning strategies which may relieve this problem, we merely introduce the simple but powerful dense reward term $R_{speed}$ to encourage agents to move forward.
>
> Note that [2] surveys various kinds of design of dense rewards for autonomous driving, including penalty for getting close to other agents [3], penalty for steering angles in ranges assumed incorrect for current command [4], penalty for deviating from the center of the lane [5], etc. As discussed in Sec 3.2, we argue that designing these fine-grained dense rewards relies too heavily on human knowledge, which may impede the learning algorithm to discover interesting interaction patterns. Besides, tuning the relative importance of these reward terms is tricky, which is not the main focus of our paper. Therefore, we abstract the necessities of autonomous agents to avoid catastrophic failures ($R_{fail}$) and introduce an easy-to-tune item ($R_{speed}$) to accelerate learning.

---

> ### Author Response · Authors · 2022-11-16
> **Response to Reviewer o4m3 (1-1)**
>
> Thank you very much for your remarkable comments. We are keenly aware that your major concern is the reproducibility of this paper since we use our internal driving simulator. In short, we are about to open-source our simulator in one month. The following are detailed responses to your concerns and questions.
>
> ---
>
> **Q1: Reproducibility - all experiments were performed on "our internal driving simulator". Can the authors give more details on this?**
>
> Please refer to the Q1 in the common response for more details.
>
> ---
>
> **Q2: The experiments were all done in this one simulator and it is unclear if the effectiveness of this strategy translates to other domains.**
>
> In this answer, we demonstrate some results on CARLA [1], an open-source 3D driving simulator. Due to time limits, we only finish some key results in $\texttt{intersection}$. CARLA TM, namely traffic manager, is an inbuilt module to control behaviors of background agents. As shown in Table-1, CARLA TM is safe but achieves the lowerest success rate, agents in CARLA TM are often stuck in the intersection. As shown in Table-2, our training framework (RARL/Ours) outperforms VRL/CoPO, which obtains the highest success rate among comparison methods in Figure 6 (in manuscript). Note that results in Table-2 align with those in Figure 6, indicating that our method can translate to other simulators.
>
> Table-1: Quantitative performance of traffic flows in $\texttt{intersection}$.
> | setting | Success ($\uparrow$) | Collision ($\downarrow$) | Other Failures ($\downarrow$)  |  Speed ($\uparrow$)  |
> | :---------: | :------: | :----------: | :----------: | :-------: |
> |  CARLA TM  |   22.0    |   **0.0**  |   **0.0**    |   32.2  |
> |  IDM (todo rerun)           |   55.2   |   37.2  |   0.0    |   59.4  |
> |  CoPO           |   77.3   |   17.8  |   5.1    |   67.2   |
> |  Ours           |   **87.1**   |   9.4   |   3.6    |   **69.4**   |
>
>
> Table-2: Driving policy's zero-shot transfer performance in $\texttt{intersection}$.
>
> | Evaluate in training background | Success ($\uparrow$) | Collision ($\downarrow$) | Other Failures ($\downarrow$) |  Speed ($\uparrow$)  |
> | :---------: | :------: | :----------: | :----------: | :-------: |
> |  VRL/CoPO  |   69.5  |   22.5  |   **7.0**  |   66.6  |
> |  RARL/Ours  |   **79.5**  |   **13.0**  |   7.5  |   **67.0**  |
>
> | &nbsp;&nbsp;&nbsp;&nbsp;&nbsp;&nbsp;&nbsp;&nbsp;&nbsp;&nbsp; Evaluate in CARLA TM &nbsp;&nbsp;&nbsp;&nbsp;&nbsp;&nbsp;&nbsp;&nbsp;&nbsp; | Success ($\uparrow$) | Collision ($\downarrow$) | Other Failures ($\downarrow$) |  Speed ($\uparrow$)  |
> | :---------: | :------: | :----------: | :----------: | :-------: |
> |  VRL/CoPO  |   49.5  |   38.5  |   **11.0**  |   59.4  |
> |  RARL/Ours |   **61.5**  |   **24.0**  |   14.5  |   **61.8**  |
>
> | &nbsp;&nbsp;&nbsp;&nbsp;&nbsp;&nbsp;&nbsp;&nbsp;&nbsp;&nbsp;&nbsp;&nbsp;&nbsp;&nbsp;&nbsp;&nbsp; Evaluate in IDM &nbsp;&nbsp;&nbsp;&nbsp;&nbsp;&nbsp;&nbsp;&nbsp;&nbsp;&nbsp;&nbsp;&nbsp;&nbsp;&nbsp;&nbsp; | Success ($\uparrow$) | Collision ($\downarrow$) | Other Failures ($\downarrow$) |  Speed ($\uparrow$)  |
> | :---------: | :------: | :----------: | :----------: | :-------: |
> |  VRL/CoPO  |   64.5  |   21.0  |   13.0  |   61.6  |
> |  RARL/Ours |   **71.5**  |   **16.5**  |   **12.0**  |   **63.9**  |

---

> ### Author Response · Authors · 2022-12-08
> **Looking forward to further discussions**
>
> Dear Reviewer,
>
> We hope that our response can address your concerns. As the deadline for the discussion period is approaching, we would appreciate it if you could let us know whether there are any further questions about the paper or the response. We are looking forward to further discussions.
>
> Best wishes, Authors

---

### Official Review · Reviewer_hLos · 2022-10-26

**Confidence:** 3
**Correctness:** 3
**Technical Novelty And Significance:** 3
**Empirical Novelty And Significance:** 3
**Recommendation:** 6

**Clarity, Quality, Novelty And Reproducibility:**

Clarity: Great; the paper is well written

Quality: Good

Originality: Good

Reproducibility: the method seems to extend upon prior work CoPO which is open-sourced; it hence seems to be reproducible based on information provided in the paper.

**Strength And Weaknesses:**

## Strengths
+ The paper is well-written and the presentation is strong.
+ The proposed method demonstrates convincing empirical results.
+ The paper proposes a novel way to adversarially train driving policies in traffic flow, and actually demonstrates that agents trained with the proposed method drive (transfers) better, which is a big plus.

## Weaknesses
- I can’t seem to find sample complexity information on traffic flow training in the paper. I am curious to see whether exposing SVO improves/decreases sample complexity with respect to the metrics in Table 1, and how it compares to the baseline CoPO.
- Table 1 compares how the proposed method impedes the driving policy. I am curious to see whether this result generalizes across different driving policies/architectures.
- I would like to see an ablation that justifies the “misunderstanding” part of the claim/story. For example, an ablation in which the driving policy also sees the SVO of the nearby traffic flow.
- The adversarial training currently strictly comes from the adversarial agent. Since the background traffic already coordinates with each other, I am wondering why not directly alternate between training between the driving policy and background traffic flow, and have the traffic flow perform multi-agent coordinated attacks to the driving policy.



**Summary Of The Paper:**

This paper presents a novel adversarial framework to train driving policies under learned traffic flow in simulation. It first extends traffic flow learned with reinforcement learning by exposing each other’s intrinsic social value orientations (SVO). Next, it uses this exposed SVO to train adversarial agents as well as a policy. In particular, the traffic flow accesses the driving policy’s SVO but not vice versa (misunderstanding). Experiments show that 1. The new traffic flow with exposed SVO to each other learns better behavior 2. The proposed adversarial training effectively reduces driving policy’s performance and 3. Driving policy has better zero-shot transferability across different traffic flows when trained with the proposed method.


**Summary Of The Review:**

The paper presents a novel framework to train driving policies under coordinated traffic flow. Experiments are thorough and demonstrate **constructive** usefulness of the presented approach. I personally think this paper is worth acceptance.

---

> ### Author Response · Authors · 2022-11-16
> **Response to Reviewer hLos (1-2)**
>
>
> **Q3: I would like to see an ablation that justifies the “misunderstanding” part of the claim/story. For example, an ablation in which the driving policy also sees the SVO of the nearby traffic flow.**
>
> In this answer, we demonstrate results on adding an ablation where the driving policy can see the SVO of the nearby traffic flow. Due to time limits, we only finish some key results in $\texttt{intersection}$.
>
> Before analyzing the following table, we want to clarify two things. First, Table 1 reveals that driving policies trained with various traffic flows (including IDM, FLOW, CoPO, and Ours wo Adv) are impeded by the adversarial policy which misunderstands our traffic flow. This justifies that misunderstanding-based adversary is effective. Second, in this work, we aim to obtain a general-purpose driving policy that can transfer to unseen traffic flows. Most prior works on training single-agent driving policy focus on maximizing self-interested reward function and do not take communication into consideration. Therefore, the driving policy is egoistic ($c_\pi = 0$) and cannot access other agents' SVOs.
>
> In the following table, VRL/Ours and RARL/Ours train $\pi$ with our traffic flow without and with adversary respectively. We add an alation VRL-SVO/Ours which trains $\pi$ with our traffic flow without adversary and can see the SVO of the nearby traffic flow. Although background agents in IDM and FLOW have no SVOs, we assume they are egoistic. The comparison between VRL/Ours and VRL-SVO/Ours shows that knowing others' SVOs improves the performance of driving policy, even in unseen environments. This result aligns with those in Figure 2 and Table 4 in appendix. Besides, when interacting with our traffic flow, adversarial agent also impedes VRL-SVO/Ours, indicating that misunderstanding can be utilized to apply adversary.
>
> Table: The success rates of driving policies in $\texttt{intersection}$.
> | Methods | IDM | FLOW | CoPO |  FailMaker   |   Ours wo Adv  |   Ours  |
> | :---------: | :------: | :----------: | :----------: | :-------: |  :-------: |  :-------: |
> |  VRL/Ours      |   69.0  |   79.0  |   74.5    |   51.5  |   87.0  |   85.0  |
> |  RARL/Ours     |   71.5  |   83.5  |   79.5    |   51.0  |   85.5  |   86.0  |
> |  VRL-SVO/Ours  |   74.5  |   82.5  |   79.0    |   55.0  |   90.5  |   87.5  |
>
> ---
>
> **Q4: The adversarial training currently strictly comes from the adversarial agent. Since the background traffic already coordinates with each other, I am wondering why not directly alternate between training between the driving policy and background traffic flow, and have the traffic flow perform multi-agent coordinated attacks to the driving policy.**
>
> We appreciate how deeply you have thought about this problem. The reason is as follows.
>
> Before going deep, we provide some observations here. First, existing adversarial attacks rely on background agents to deliberately induce driving policy failures. In other words, background agents are trained to collide with the driving agent. Second, during training the coordinated traffic flow, collisions between agents are inevitable. Third, The convergence rate of multi-agent training is slower than that of single-agent.
>
> Given these observations, we can find that if we do so, adversaries come from two aspects, i.e., coordinated attack and collision between agents (although this background traffic flow is not willing to collide with the driving agent). Besides, when the driving agent is well-trained, the background traffic flow may still continually collide with the driving agent. As one can see, this training pipeline works similarly to existing adversarial attack methods and the effect of coordinated attack will be submerged. As discussed in Sec 1, the transferability of driving policy cannot be improved with this setting.
>
> ---
>
> **References:**
>
> [1] Zhenghao Peng, Quanyi Li, Ka Ming Hui, Chunxiao Liu, and Bolei Zhou. Learning to simulate self-driven particles system with coordinated policy optimization. Advances in Neural Information Processing Systems, 34:10784–10797, 2021.

---

> ### Author Response · Authors · 2022-11-16
> **Response to Reviewer hLos (1-1)**
>
>
> Thank you very much for the constructive feedback. We appreciate that you find our paper convincing, novel, and well-written. We address additional comments below.
>
> ---
>
> **Q1: I can’t seem to find sample complexity information on traffic flow training in the paper. I am curious to see whether exposing SVO improves/decreases sample complexity with respect to the metrics in Table 1, and how it compares to the baseline CoPO.**
>
> We don't exactly grasp the meaning of "sample complexity". From our perspective, the changes in metrics during training could reveal the sample complexity. We train 1000 episodes for each method (more details can be found in Q2 of the common response) and success rates of the first 200 episodes are shown in the following table (complete results will be added in the next version). As one can see, during the first 100 episodes, CoPO outperforms Ours in most scenarios, indicating that exposing SVO is harder to train at the very beginning. After that, Ours outperforms CoPO consistently, implying that exposing SVO decreases sample complexity and achieves a higher coordinate level.
>
>
> Table: The success rates of CoPO and our traffic flow during training.
> | Settings |  0 |  20 |  40 |  60 |  80 |  100 |  120 |  140 |  160 |  180 |  200 |
> | :---------: | :--: | :--: | :--: | :--: |  :--: | :--: | :--: | :--: | :--: | :--: | :--: |
> $\texttt{intersection}$ (CoPO) |   0.0 |  0.7 |  12.8 |  23.3 |  32.8 |  39.3 |  44.6 |  48.7 |  51.9 |  53.4 |  55.1 |
> $\texttt{intersection}$ (Ours) |   0.0 |  0.7 |  10.9 |  20.9 |  31.7 |  40.0 |  45.9 |  50.6 |  53.8 |  55.7 |  58.5 |
> &nbsp;
> $\texttt{bottleneck}$ (CoPO) |   3.1 |  2.2 |  25.0 |  25.7 |  27.9 |  34.7 |  40.4 |  43.9 |  47.6 |  50.2 |  51.6 |
> $\texttt{bottleneck}$ (Ours) |   2.4 |  3.0 |  20.3 |  23.4 |  31.1 |  38.2 |  43.2 |  47.3 |  49.4 |  50.9 |  53.6 |
> &nbsp;
> $\texttt{merge}$ (CoPO) |   4.1 |  1.8 |  24.7 |  36.7 |  40.9 |  43.7 |  46.6 |  48.4 |  50.0 |  51.8 |  53.1 |
> $\texttt{merge}$ (Ours) |   2.3 |  1.5 |  27.9 |  39.9 |  41.7 |  45.3 |  49.0 |  51.2 |  52.9 |  54.9 |  57.0 |
> &nbsp;
> $\texttt{roundabout}$ (CoPO) |   0.0 |  1.1 |  22.5 |  32.5 |  38.5 |  43.1 |  47.8 |  51.2 |  53.6 |  55.5 |  57.4 |
> $\texttt{roundabout}$ (Ours) |   0.0 |  1.1 |  18.3 |  26.9 |  38.5 |  43.6 |  48.0 |  52.2 |  54.6 |  56.2 |  58.0 |
>
> ---
>
> **Q2: Table 1 compares how the proposed method impedes the driving policy. I am curious to see whether this result generalizes across different driving policies/architectures.**
>
> In our humble opinion, different policy architectures can achieve similar results in Table 1 if the observation space contains the same information. For Table 1, "similar" means that the adversarial agent can degrade the success rate of driving policy across different architectures.
>
> The spurious adversarial agent impedes the driving policy by misunderstanding the traffic flow. On the one hand, for training such an agent, the effect of $c_\beta$ (the output of adversarial policy) towards the traffic flow must be captured, therefore, the input of adversarial policy must contain information about nearby traffic flow. On the other hand, RL has achieved widespread success in various domains including visual games, recommender systems, and robotics with diverse network architectures. We believe that model-free RL algorithms are relatively agnostic to policy architectures. Given similar input information, policies can achieve similar outcomes.
>
> Indirect support is as follows. We take Figure 2 as an example, in which CoPO outperforms FLOW in almost all aspects. This is the evidence of the claim "incorporating SVO into Independent Policy Learning produces coordinated behaviors", which is also an important contribution of the CoPO paper [1]. They use PPO and MLP with Lidar-based observations while we use SAC and a VectorNet-like architecture with vectorized observations. Although quite dissimilar, this claim still holds. Therefore, we infer that, for Table 1, the claim "the adversarial agent degrades the success rate of driving policy across different architectures" is true.
>
> Honestly, running a bunch of experiments with a new policy architecture is quite challenging and time-consuming in practice. However, generalization across different policy architectures for RL-oriented tasks is a significant and valuable idea, especially for autonomous driving. We will explore this interesting problem in the future.

---

> ### Author Response · Authors · 2022-12-08
> **Looking forward to further discussions**
>
> Dear Reviewer,
>
> We hope that our response can address your concerns. As the deadline for the discussion period is approaching, we would appreciate it if you could let us know whether there are any further questions about the paper or the response. We are looking forward to further discussions.
>
> Best wishes, Authors

---

### Official Review · Reviewer_7Hpn · 2022-11-02

**Confidence:** 2
**Correctness:** 3
**Technical Novelty And Significance:** 2
**Empirical Novelty And Significance:** 2
**Recommendation:** 5

**Clarity, Quality, Novelty And Reproducibility:**

I think the authors can improve the clarity of the submission substantially. Specifically, I'm having trouble identifying the main result that the authors achieve using their method. I'm also somewhat confused about Section 3.3 - is there a reason the authors use this architecture compared to prior work? Can the authors define formally what it means for their framework to be "misunderstanding-based"? Once I have a better sense of the main takeaway of the paper, I will be more confident in commenting on the originality of the paper.

**Strength And Weaknesses:**

One strength of this paper is that it addresses multiple impactful problems in self-driving simulation. Generally speaking, adversarial NPCs exhibit unrealistic aggressive behavior. The idea of controlling the extent to which the NPCs optimize for the reward of nearby agents seems like a reasonable way to parameterize the space of plausible vehicle behaviors that an AV may interact with on real roads.

That being said, the main weakness for me reading this paper was that I found it hard to understand precisely what the authors claim as a contribution in this paper. My sense is that the main contribution is the use of $c_\beta$ as the variable to adversarially optimize. However, my sense is another contribution is maybe the construction of NPCs that have such a $c_\beta$? Or is the main contribution the policy that is ultimately trained using Algorithm 1 which they show can be transferred to new test-time simulators with out-of-distribution NPC controllers? I'm not seeing yet what is novel about Algorithm 1 - do the authors define what is different between "misunderstanding-based adversarial learning" and standard adversarial learning?

I list my other minor edits and questions below:

- state space - no notion of length or width?

- "Although IPL is prone to generate egoistic suboptimal behaviors" - is there a citation for this claim?

- Figure 3 - do the authors compare against the baseline of training the policy on a hard-coded uniform distribution for $c_\beta$ instead of the adversarially optimized distribution that they find for $c_\beta$?

- Section 5.1 - clarify why would the initial poses being close together imply that it's harder for the agents to coordinate?

- Figure 5 - I think it's easier to read if the same coordinate frame is used for each column

- Figure 6 - I find this figure difficult to read. I think it would be better if the authors use the same colormap for all grid cells instead of using blue only for the diagonal.

- Do the authors have videos they can share? It's hard to understand the effect of different $c_\beta$ from frame shots.

- Section 5.3 - I'm not sure what it means for the evaluation environment to be "IID with training environment"?

- Figures and Tables - I feel "ours" is being used to sometimes denote NPC policies, sometimes ego policies, and sometimes training environments. Is there a way the authors could disambiguate each of these contributions?

**Summary Of The Paper:**

The authors build a traffic simulator in which each agent drives in order to maximize its reward as well as the mean reward of nearby agents. The extent to which the vehicle optimizes for the reward of other agents $c_\beta$ can be adjusted. The authors then show that if a new agent is trained in such a simulator and during training the parameter for controlling $c_\beta$ is adversarially optimized, the resultant policy transfers better than baselines to new traffic densities and test-time out-of-distribution NPC controllers (e.g. IDM).

**Summary Of The Review:**

I'm currently unclear on how to interpret some of the main claims of the paper, namely whether the authors are claiming that they have a general-purpose driving policy, or a general-purpose simulator in which robust driving policies can be trained. I also think that qualitative videos would make it much easier to judge the quality of the contribution, since most of the metrics are designed by the authors. If the authors clarify their contribution, I am willing to increase my score.

---

> ### Author Response · Authors · 2022-11-16
> **Response to Reviewer 7Hpn (1-4)**
>
>
> **References:**
>
> [1] Alexey Dosovitskiy, German Ros, Felipe Codevilla, Antonio Lopez, and Vladlen Koltun. Carla: An open urban driving simulator. In Conference on robot learning, pp. 1–16. PMLR, 2017.
>
> [2] Dian Chen, Brady Zhou, Vladlen Koltun, and Philipp Kra ̈henbu ̈hl. Learning by cheating. In Conference on Robot Learning, pp. 66–75. PMLR, 2020.
>
> [3] Cathy Wu, Abdul Rahman Kreidieh, Kanaad Parvate, Eugene Vinitsky, and Alexandre M Bayen. Flow: A modular learning framework for mixed autonomy traffic. IEEE Transactions on Robotics, 2021.
>
> [4] Zhenghao Peng, Quanyi Li, Ka Ming Hui, Chunxiao Liu, and Bolei Zhou. Learning to simulate self-driven particles system with coordinated policy optimization. Advances in Neural Information Processing Systems, 34:10784–10797, 2021.
>
> [5] Simon Suo, Sebastian Regalado, Sergio Casas, and Raquel Urtasun. Trafficsim: Learning to simulate realistic multi-agent behaviors. In Proceedings of the IEEE/CVF Conference on Computer Vision and Pattern Recognition, pp. 10400–10409, 2021.
>
> [6] Nicholas Rhinehart, Rowan McAllister, and Sergey Levine. Deep imitative models for flexible inference, planning, and control. In International Conference on Learning Representations, 2019.
>
> [7] Yilun Chen, Chiyu Dong, Praveen Palanisamy, Priyantha Mudalige, Katharina Muelling, and John M Dolan. Attention-based hierarchical deep reinforcement learning for lane change behaviors in autonomous driving. In Proceedings of the IEEE/CVF Conference on Computer Vision and Pattern Recognition Workshops, pp. 0–0, 2019b.
>
> [8]Majid Moghadam, Ali Alizadeh, Engin Tekin, and Gabriel Hugh Elkaim. An end-to-end deep reinforcement learning approach for the long-term short-term planning on the frenet space. arXiv preprint arXiv:2011.13098, 2020.
>
> [9] Panpan Cai, Yuanfu Luo, Aseem Saxena, David Hsu, and Wee Sun Lee. Lets-drive: Driving in a crowd by learning from tree search. In Proceedings of Robotics: Science and Systems, FreiburgimBreisgau, Germany, June 2019. doi: 10.15607/RSS.2019.XV.018.
>
> [10] Jianyu Chen, Bodi Yuan, and Masayoshi Tomizuka. Model-free deep reinforcement learning for urban autonomous driving. In 2019 IEEE intelligent transportation systems conference (ITSC), pp. 2765–2771. IEEE, 2019a.
>
> [11] Marin Toromanoff, Emilie Wirbel, and Fabien Moutarde. End-to-end model-free reinforcement learning for urban driving using implicit affordances. In Proceedings of the IEEE/CVF conference on computer vision and pattern recognition, pp. 7153–7162, 2020.

---

> ### Author Response · Authors · 2022-11-16
> **Response to Reviewer 7Hpn (1-3)**
>
> **Q6: I'm having trouble identifying the main result that the authors achieve using their method. Figure 6 - I find this figure difficult to read. I think it would be better if the authors use the same colormap for all grid cells instead of using blue only for the diagonal.**
>
> Thank you for pointing out this issue. We apologize for the confusion we introduce and will take your advice in the next version of this paper.
>
> ---
>
> **Q7: Section 5.3 - I'm not sure what it means for the evaluation environment to be "IID with training environment"?**
>
> We apologize for the confusion we introduce. In this paper, "IID with training environment" means that background traffic flow in evaluating is the same as in training for one driving policy. Contrastly, "OOD / zero-shot transfer" means that background traffic flow in evaluating is different from that in training. We will improve our expression in the next version.
>
> ---
>
> **Q8: Figures and Tables - I feel "ours" is being used to sometimes denote NPC policies, sometimes ego policies, and sometimes training environments. Is there a way the authors could disambiguate each of these contributions?**
>
> Thank you for pointing out this issue. We apologize for the confusion we introduce with this "ours" description. We will find out a way to disambiguate these contributions in the next version of this paper.
>
> ---
>
> **Q9: I'm also somewhat confused about Section 3.3 - is there a reason the authors use this architecture compared to prior work?**
>
> As discussed in the common response, compared to rasterized representation with architectures like ResNet, we utilize vectorized representation with a VectorNet-like architecture because of its computation and memory efficiency.
>
> The comparisons between VectorNet and our feature extraction framework are as follows. Both frameworks are hierarchical and Multi-Head Attention (MHA) is used for high-level feature extraction. The main difference is the low-level feature extraction. VectorNet uses a single PointNet to extract features from static and dynamic elements jointly. We use two DeepSets to extract features from static and dynamic elements separately.
>
> First, we use DeepSet because it is powerful to deal with homogeneous elements and has theoretical guarantees on the approximation of continuous function operating on permutation invariant elements. Second, using two networks to separately process static and dynamic elements saves GPU memory compared to using one network to jointly process static and dynamic elements. The reason is that in joint processing, dynamic and static elements need to be aligned and stacked before putting on GPU. The following example illustrates this. Suppose the size of a dynamic- and static-element tensor is `[20, 10, 7]` and `[30, 20, 6]` respectively. The input size of joint processing will be `[sum(20,30), max(10,20), max(7,6))]`, totally $50 \times 20 \times 7 = 7000$ floats. However, the maximum float of separate processing is $\max(20 \times 10 \times 7, 30 \times 20 \times 6) = 3600$, saving nearly $50\%$ floats that a GPU needs to accommodate.
>
> ---
>
> **Q10: Do the authors have videos they can share? It's hard to understand the effect of different $c_{\beta}$ from frame shots. I also think that qualitative videos would make it much easier to judge the quality of the contribution, since most of the metrics are designed by the authors.**
>
> Thank you for pointing out this issue. We agree that a video gives more intuition than frame shots. We are working on it! Once we finish it, we will notify you immediately.
>
> Last but not the least, we clarify that most metrics we use are not designed by us but are widely used in prior works. Success is used in [1,2,3,4,6,7,9]. Collision is used in [2,5,7,9]. Off Road is used in [6,7]. Off Route is our design, but [10,11] uses it as a reward term. Wrong Lane is used in [6]. Efficiency is used in [3,7,8,9]. We will add these references in the next version.

---

> ### Author Response · Authors · 2022-11-16
> **Response to Reviewer 7Hpn (1-2)**
>
> **Q2: Can the authors define formally what it means for their framework to be "misunderstanding-based"? I'm not seeing yet what is novel about Algorithm 1 - do the authors define what is different between "misunderstanding-based adversarial learning" and standard adversarial learning?**
>
> In this answer, we simplify the notation about MDP and focus on different optimization objectives.
>
> For standard adversarial learning:
> $$
> \max_{\pi} \ \mathbb{E}_{\tau \sim \pi, \beta} \ [R(\tau)]
> $$
>
> $$
> s.t. \quad  \max_{\beta} \  \mathbb{E}_{\tau \sim \pi, \beta} \ [\cos(d) R_b(\tau) + \sin(d)R(\tau)], \quad d \in [-\frac{\pi}{2}, 0)
> $$
> where $R$ is the reward function of POMDP $M$ (in Sec 4.1), $R_b$ is the reward function of adversarial background agents. The only difference between $R_b$ and reward function of POSG (in Sec 3.1) is that $R_b$ does not penalize collision between agents. Background policy $\beta$ is optimized to degrade the performance of driving policy $\pi$ and keep natural behaviors like keeping in the right lane and driving efficiently. When $d = -\frac{\pi}{2}$, above adversarial learning is a standard minimax problem.
>
> For misunderstanding-based adversarial learning:
>
> $$
> \max_{\pi} \ \mathbb{E}_{\tau \sim \pi, \beta, \pi_c} \ [R(\tau)]
> $$
>
> $$
> s.t. \quad  \max_{\pi_c} \  \mathbb{E}_{\tau \sim \pi, \beta, \pi_c} \ [-R(\tau)]
> $$
>
> where background policy $\beta$ is optimized to coordinate (equation 2 in manuscript):
>
> $$
> s.t. \quad  \max_{\beta} \  \mathbb{E}_{\tau \sim \beta} \ [\cos(c) R(\tau) + \sin(c)R_S(\tau)], \quad c \in [0, \frac{\pi}{2}]
> $$
>
> The main difference between standard and misunderstanding-based adversarial learning is the objective of $\beta$. In standard adversarial learning, $\beta$ controls background agents to attack the driving agent. Background agents know exactly which one is the driving agent. While in misunderstanding-based adversarial learning, background agents aim to coordinate with each other, including the driving agent. A background agent cannot distinguish which surrounding agent is the driving agent. Therefore, the spurious agent which produces $c_\beta$ applies adversary from the perspective of driving agent and this is the reason why in Algorithm 1 the spurious agent and driving policy take the same observation.
>
> Algorithm 1 exhibits minimax optimization between $\pi$ and $\pi_c$, which works similar to standard adversarial learning with $d = -\frac{\pi}{2}$. However, as discussed below, the novelty of Algorithm 1 comes from the adversary variable $c_\beta$ and the coordinated background policy $\beta$. We will add this discussion to the next version of our paper.
>
> ---
>
> **Q3: I'm currently unclear on how to interpret some of the main claims of the paper, namely whether the authors are claiming that they have a general-purpose driving policy, or a general-purpose simulator in which robust driving policies can be trained.**
>
> We claim a general-purpose driving policy and want this driving policy to be robust and transferable. A general-purpose driving agent means that its SVO is 0 and it cannot access other agents' SVOs. This aligns with settings in previous single-agent researches for autonomous driving. Under this assumption, we design a specific-purpose background traffic flow where agents can communicate their SVOs.
>
> ---
>
> **Q4: Figure 3 - do the authors compare against the baseline of training the policy on a hard-coded uniform distribution for $c_{\beta}$ instead of the adversarially optimized distribution that they find for $c_{\beta}$?**
>
> Thank you for your idea. We are working on it and will let you know once we finish it.
>
> ---
>
> **Q5: Section 5.1 - clarify why would the initial poses being close together imply that it's harder for the agents to coordinate?**
>
> The number of agents reveals the distance of initial poses. In the following table, we train our traffic flow with 8 to 20 agents and evaluate success rates from 2 to 20 agents respectively. We find that success rates decrease with the increase of $n$, indicating that it is harder for agents to coordinate in denser traffic flow. Besides, the available initial poses are close together compared to the other three scenarios, we will add a figure to demonstrate this in the next version.
>
> Table: The change in success rates with the number of agents.
> | Num Agents | $\texttt{Intersection}$ | $\texttt{Bottleneck}$ | $\texttt{Merge}$ |  $\texttt{Roundabout}$   |
> | :---------: | :------: | :----------: | :----------: | :-------: |
> |  $n=2$   |   99.2  |   99.6  |   100.0  |   99.4  |
> |  $n=4$   |   98.5  |   99.2  |   99.8   |   98.0  |
> |  $n=8$   |   96.3  |   97.2  |   99.0   |   97.5  |
> |  $n=12$  |   93.4  |   95.3  |   96.3   |   93.9  |
> |  $n=16$  |   89.3  |   90.4  |   89.8   |   85.9  |
> |  $n=20$  |   86.9  |   83.4  |   83.1   |   84.6  |

---

> ### Author Response · Authors · 2022-11-16
> **Response to Reviewer 7Hpn (1-1)**
>
> Thank you very much for providing such a detailed review. We apologize for the confusion we introduce in the manuscript and address your concerns below. We are keenly aware that your major concerns are the main contribution of our paper and the difference between misunderstanding-based adversarial learning and standard adversarial learning. The first three answers address your major concerns, followed by other answers.
>
> ---
>
> **Q1: the main weakness for me reading this paper was that I found it hard to understand precisely what the authors claim as a contribution in this paper. My sense is that the main contribution is the use of $c_{\beta}$ as the variable to adversarially optimize. However, my sense is another contribution is maybe the construction of NPCs that have such a $c_{\beta}$? Or is the main contribution the policy that is ultimately trained using Algorithm 1 which they show can be transferred to new test-time simulators with out-of-distribution NPC controllers?**
>
> Your sense is right. Our main contribution is the use of $c_{\beta}$ to apply adversarial training in dense traffic flow. Trained with such an approach, the driving policy can be transferred to new test-time simulators with out-of-distribution NPC controllers. To our best knowledge, our paper is the first work to investigate driving policy's transferability across different background policies. Related works are can be divided into two categories as discussed in Sec 2. Existing adversarial attack methods demonstrate that driving policy can resist disturbances produced by adversarial background agents after adversarial training. However, whether the driving policy can act robustly against unseen background agents remains an open problem. Another line of work focus on building coordinated dense traffic flows, while the interactions between driving policy and traffic flows are overlooked.
>
> In addition, we want to clarify the phraseology "the construction of NPCs that have such a $c_{\beta}$". NPC or background agent in our traffic flow has their own SVO and recognizes others' SVOs. The SVOs of all agents are denoted as $c_0, c_1, \dots, c_{n-1}$. Our traffic flow is coordinated by incorporating SVO and allowing agents to communicate their SVOs. For a general-purpose driving agent, its SVO $c_{\pi}$ is always 0, since existing single-agent algorithms are fully self-interested. When the driving agent is deployed to interact with this traffic flow, background agents need to know the SVO of the driving agent $c_\beta$. If $c_{\beta} = c_\pi$, the traffic flow coordinates with the driving agent. If $c_{\beta} \neq c_\pi$, coordination is broken and we say that "misunderstanding" emerges. We hence introduce a spurious agent to adversarially select $c_\beta$ and formulate a misunderstanding-based adversarial training framework.

---

> ### Author Response · Authors · 2022-12-08
> **Looking forward to further discussions**
>
> Dear Reviewer,
>
> We hope that our response can address your concerns. As the deadline for the discussion period is approaching, we would appreciate it if you could let us know whether there are any further questions about the paper or the response. We are looking forward to further discussions.
>
> Best wishes, Authors

---

### Author Response · Authors · 2022-11-16
**Common Response to All Reviewers (1-2)**


**Q1: Details of our internal driving simulator. (continue)**

#### *done condition*

In our simulator, one agent is done if it reaches its destination, encounters catastrophic failures, or survives until timeout. Once an agent is done, it will be removed from the scenario. When all foreground agents are done, this episode ends. Catastrophic failures include collision with other agents, deviation from drivable area, driving too far from global path, and crashing into wrong lane. The maximum steps for one episode are $t_{max}$ and when an agent survives $t_{max}$ steps in the environment, we call it "timeout". In this paper, $t_{max}=100$.

An agent is marked as success only when it passes the interaction zone. Note that we name each scenario with its interaction zone. For instance, the interaction zone in $\texttt{bottleneck}$ is the bottleneck. We will add a figure to visualize interaction zones in the next version of this paper.

---

**Q2: Details of training and evaluating settings.**

#### *details of training algorithm*

We use PyTorch 1.9.1 to construct policy architectures and use Adam [5] to optimize policies. All models are trained with one Nvidia RTX 3090. We utilize Soft-Actor-Critic (SAC) [6] for single-agent training and its independent version (ISAC) for multi-agent training. We follow https://github.com/pranz24/pytorch-soft-actor-critic and detailed parameters are shown in Table 2 (in Appendix). Note that we use precisely the same parameters for both single- and multi-agent policy learning.

#### *details of training settings*

We use 16 workers with different seeds to sample experiences in parallel across four scenarios including $\texttt{intersection}$, $\texttt{bottleneck}$, $\texttt{merge}$, and $\texttt{roundabout}$. Each scenario is assigned 4 workers. There are two different settings in each scenario. One setting is that the SVOs of all agents are the same, i.e., $c_i = c \sim \mathcal{U}(0,\frac{\pi}{2})$. Another setting is that each agent has a distinct SVO, i.e., $c_i \sim \mathcal{U}(0,\frac{\pi}{2})$. Each setting is assigned 2 workers. For each scenario, we randomly place 8 to 20 agents.

Since we use Ray [7] to implement parallel sampling, the training procedure and sampling procedure are in different processes, and passing network parameters frequently among processes is costly. Therefore, instead of sampling one experience and updating parameters one time (which is a common choice for off-policy algorithms with one process/worker), we first run 5 episodes to sample $N_e \approx 5 \times 16 \times 100$ experiences, and then update parameters $N_e$ times. We iterate this procedure 200 times (totally around 16000000 steps).

#### *details of evaluating settings*

For evaluating various methods fairly, we offline generate 200 cases (with different global paths, initial poses, and SVOs) for each scenario. Each case contains 20 agents. For each method/setting, we run these cases 5 times with different seeds and average the metrics. For all results except for Figure 3, 7-10, each agent has a distinct SVO. For Figure 3, 7-10, all agents share the same SVO in each data point and We set the $\sin$ of SVO from 0 to 1 at interval 0.1.

---

**References:**

[1] Ming Liang, Bin Yang, Rui Hu, Yun Chen, Renjie Liao, Song Feng, and Raquel Urtasun. Learning lane graph representations for motion forecasting. In European Conference on Computer Vision, pp. 541–556. Springer, 2020.

[2] Hang Zhao, Jiyang Gao, Tian Lan, Chen Sun, Ben Sapp, Balakrishnan Varadarajan, Yue Shen, Yi Shen, Yuning Chai, Cordelia Schmid, et al. Tnt: Target-driven trajectory prediction. In Conference on Robot Learning, pp. 895–904. PMLR, 2021.

[3] Junru Gu, Chen Sun, and Hang Zhao. Densetnt: End-to-end trajectory prediction from dense goal sets. In Proceedings of the IEEE/CVF International Conference on Computer Vision, pp. 15303– 15312, 2021.

[4] Jiyang Gao, Chen Sun, Hang Zhao, Yi Shen, Dragomir Anguelov, Congcong Li, and Cordelia Schmid. Vectornet: Encoding hd maps and agent dynamics from vectorized representation. In Proceedings of the IEEE/CVF Conference on Computer Vision and Pattern Recognition, pp. 11525–11533, 2020.

[5] Diederik P Kingma and Jimmy Ba. Adam: A method for stochastic optimization. In ICLR (Poster), 2015.

[6] Tuomas Haarnoja, Aurick Zhou, Pieter Abbeel, and Sergey Levine. Soft actor-critic: Off-policy maximum entropy deep reinforcement learning with a stochastic actor. In International conference on machine learning, pp. 1861–1870. PMLR, 2018.

[7] Philipp Moritz, Robert Nishihara, Stephanie Wang, Alexey Tumanov, Richard Liaw, Eric Liang, Melih Elibol, Zongheng Yang, William Paul, Michael I Jordan, et al. Ray: A distributed framework for emerging {AI} applications. In 13th USENIX Symposium on Operating Systems Design and Implementation (OSDI 18), pp. 561–577, 2018.

---

### Author Response · Authors · 2022-11-16
**Common Response to All Reviewers (1-1)**

We sincerely thank all the reviewers for your insightful comments. In this common response, we clarify the main concerns on the reproducibility of this work. In short, we are about to open-source our simulator in one month. The following are details of our simulator, followed by training and evaluating implementation details.

---

**Q1: Details of our internal driving simulator.**

Our internal driving simulator is 2D and aims to investigate single- and multi-agent driving behaviors, especially in dense traffic flows. Inspired by the trajectory prediction community [1,2,3], our simulator utilizes sparse (vectorized) representation to capture the structural information of high-definition maps and agents. Compared to rasterized encoding which rasterizes the HD map elements together with agents into an image, vectorized representation is computation- and memory-efficient [4]. The critical components of our simulator are built on top of this vectorized representation.

For designing an RL-oriented simulator, there are three critical components including scenario initialization (`env.reset`), step forward (`env.step`), and done condition (`done`). We provide a python-style pseudo code to illustrate how to run one episode with these components as below.

```python
def run_one_episode(env):
    state = env.reset()
    while True:
        action = env.action_space.sample()
        next_state, reward, done, info = env.step(action)
        if done:
            break
        state = next_state
    return
```

#### *scenario initialization*

We choose one scenario from $\texttt{intersection}$, $\texttt{bottleneck}$, $\texttt{merge}$, and $\texttt{roundabout}$ and load the pre-built vectorized map for this scenario. After that, we assign global path, initial pose, and SVO for each agent in the scenario. A vectorized map contains two parts including the centerline and sideline. Each part is a 3D `numpy.ndarray` which contains different elements. Each element contains a sequence of points. Each point $v$ is a vector $[p, l]$ where $p = (x, y, \theta)$ is the pose and $l$ represents the lane width (for sideline, $l$ is always $0$). The average distance of adjacent points is $2.0m$. Given this map, we utilize `networkx` to build a graph $\mathcal{G}$ on top of centerline where each element is a node of $\mathcal{G}$ and an edge exists only when two elements are connected end to end. For each scenario, we manually pick up two bunch of points as initial and terminational poses respectively. For each agent, we randomly select an initial and terminal pose ($p_{initial}$ and $p_{terminal}$) and use A* algorithm to search a list of points from $p_{initial}$ to $p_{terminal}$ on $\mathcal{G}$. This list of points is the global path of the agent, in which the first point is the initial pose. Finally, we assign a SVO $c \in [0,\frac{\pi}{2}]$ to this agent. For all agents in the scenario, the above procedure is repeated.

Before going into the next component, we further clarify what the term "agent" means in our paper. Agents can be divided into two categories: foreground agents and background agents. Background agents are NPCs that have unchanged policies like IDM and learned NNs. Background agents are part of the environment. For a single-agent environment, there is only one single foreground agent (driving agent). For a multi-agent environment, there exist multiple foreground agents. "Foreground agent" is exactly the meaning of "agent" in RL community. Currently, when there are $n$ vehicles in our simulator, the number of foreground and background agents is ($1, n-1$) for single-agent settings and ($n, 0$) for multi-agent settings.

#### *step forward*

We use bicycle model as the vehicle dynamic model, where the inputs of the model are acceleration $a$ and steer $\delta$ and the state is $(x, y, \theta, v)$. To guarantee that the agent will not exceed its maximum speed $v_{max} = 6m/s$ substantially, we introduce a PID controller ($K_p=1.0$, $K_i=0.01$, $K_d=0.05$) to regulate $a$ given the reference speed $v_r$ and current speed $v$. Therefore, the action is $(v_r, \delta)$ where $v_r \in [0, v_{max}]$ and $\delta \in [-45^{\circ}, 45^{\circ}]$. As explained in Sec 3.2, the state and observation space contains a collection of static and dynamic elements and is vectorized, the dimension of state and observation space is inherently not fixed. The length of static elements is not fixed and has no upper bound. The upper bound of dynamic elements' length is 10.

---

### Author Response · Authors · 2022-11-18
**Common Response to All Reviewers (Paper Revised)**

We have updated our rebuttal revision. We correct all minor issues that all reviewers mentioned. Please let us know if you have any remaining concerns or questions. Best wishes!

---

### Decision · Program_Chairs · 2023-01-20

**Decision:**

Reject

**Justification For Why Not Higher Score:**

Although this is exciting work, the reviewers will be in a better position to assess the strength of the simulated results once the simulator is accessible and can be validated by the community.

**Justification For Why Not Lower Score:**

N/A

**Metareview: Summary, Strengths And Weaknesses:**

This paper considers the problem of designing adversarial agents to improve learned policies of AI driving agents. In general this is a neat idea, although the reviewers expressed some concerns that it is not clear what is the actual novel contribution here. More generally, the reviewers expressed concerns regarding the clarity of the current paper, as well as difficulty understanding the quality of the experimental results given the closed-source nature of the simulator. I appreciate the authors will be open-sourcing their simulator soon - this should make it easier for reviewers to understand the empirical significance of this work.